# Histamine $H_1$ receptors in dentate gyrus-projecting cholinergic neurons of the medial septum suppress contextual fear retrieval in mice

Li Cheng[1,2], Ling Xiao[1,2], Wenkai Lin [1,2], Minzhu Li[1], Jiaying Liu[1], Xiaoyun Qiu[1], Menghan Li[1], Yanrong Zheng[1], Cenglin Xu [1], Yi Wang [1] & Zhong Chen [1]✉

Fear memory is essential for survival and adaptation, yet excessive fear memories can lead to emotional disabilities and mental disorders. Despite previous researches have indicated that histamine $H_1$ receptor ($H_1$R) exerts critical and intricate effects on fear memory, the role of $H_1$R is still not clarified. Here, we show that deletion of $H_1$R gene in medial septum (MS) but not other cholinergic neurons selectively enhances contextual fear memory without affecting cued memory by differentially activating the dentate gyrus (DG) neurons in mice. $H_1$R in cholinergic neurons mediates the contextual fear retrieval rather than consolidation by decreasing acetylcholine release pattern in DG. Furthermore, selective knockdown of $H_1$R in the MS is sufficient to enhance contextual fear memory by manipulating the retrieval-induced neurons in DG. Our results suggest that $H_1$R in MS cholinergic neurons is critical for contextual fear retrieval, and could be a potential therapeutic target for individuals with fear-related disorders.

Excessive fear memories can lead to emotional disabilities and mental disorders, such as anxiety disorders and post-traumatic stress disorder (PTSD)[1,2]. Currently, psychotherapy and medication are considered as first-line therapies in the treatment of fear-related disorders[3,4]. However, extinction-based exposure therapy (ET) is emotionally challenging and insufficiently effective because approximately 35% of patients fail to respond to treatment[5–7]. Moreover, the combination of ET and pharmacological treatment, such as selective serotonin reuptake inhibitors (SSRIs), shows little further advantage for patients[5,8]. Thus, further elucidation for the mechanism of fear memory and investigation into new classes of drugs with the potential to treat fear-related disorders are warranted.

Histamine is a relatively moderate neurotransmitter, yet it plays a vital role in many pathophysiological processes, including wakefulness-sleep, feeding, learning, and memory, primarily through the

histamine $H_1$ receptor ($H_1$R)[9,10]. A large number of studies have highlighted a close link between histamine or its receptors and fear memory. Studies using histidine decarboxylase knockout (HDC-KO) mice have indicated that long-term histamine deficiency facilitates contextual fear memory, which may result from increased hippocampal CA1 LTP and presynaptic glutamate release[11]. Conversely, histamine depletion obtained by using intralateral ventricle administration of α-fluoromethylhistidine (α-FMH, a suicide inhibitor of HDC) impairs long-term but not short-term memory on one-trial step-down inhibitory avoidance task[12]. Histamine $H_1$ receptor gene knockout ($H_1$R-KO) mice exhibit a pronounced enhancement in auditory and contextual fear conditioning tests when compared to the respective wild-type mice[13], whereas the consolidation and expression of conditioned fear have been shown to be impaired by peroral treatment of the first-generation antihistamine diphenhydramine[14]. It

[1]Key Laboratory of Neuropharmacology and Translational Medicine of Zhejiang Province, School of Pharmaceutical Sciences, The First Affiliated Hospital of Zhejiang Chinese Medical University (Zhejiang Provincial Hospital of Chinese Medicine), Zhejiang Chinese Medical University, Hangzhou, China. [2]These authors contributed equally: Li Cheng, Ling Xiao, Wenkai Lin. ✉e-mail: chenzhong@zju.edu.cn

implies that $H_1R$ may play an intricate role in the regulation of fear memory. We have previously demonstrated the brain region- and cell type-specific functions of $H_1R$ is involved in the pathogenesis of schizophrenia[15]. The contradictory results make further research into the function of $H_1R$ in certain type of neuron and certain brain region in fear memory worthwhile.

Here, we specifically deleted *Hrh1* in cholinergic neurons using the Cre-LoxP system and found that *ChAT-Cre;Hrh1$^{fl/fl}$* mice displayed enhanced contextual fear memory and normal cued fear memory. Moreover, we combined *Hrh1* restoration and RNA interference in medial septal (MS) cholinergic neurons to dissect the potentially differential regions of control and *ChAT-Cre;Hrh1$^{fl/fl}$* mice in contextual fear memory. The role of $H_1R$ in MS cholinergic neurons in contextual fear retrieval but not consolidation was also explored by modulating the retrieval-induced neurons in the dentate gyrus (DG). Our results show that $H_1R$ in MS cholinergic neurons critically contributes to the retrieval of contextual fear memory. This is a promising target for the alteration of fear memory, and may be of therapeutic importance in the treatment of fear-related disorders.

## Results

### Decreased histamine $H_1R$ expression in cholinergic neurons selectively enhances contextual fear memory

Given the essential role of cholinergic system in regulation of cognitive functions[16–19], we crossed the *ChAT-Cre* mice with *Hrh1$^{fl/fl}$* mice to induce *Hrh1* (the $H_1R$ gene) deletion in cholinergic neurons. In a previous study, we have shown that the *ChAT-Cre;Hrh1$^{fl/fl}$* mice exhibited similar functional capacity for olfactory function, motor coordination, body growth rate, body temperature, pain sensitivity, and normal locomotor activity when compared with littermates[15]. To investigate the role of $H_1R$ in the cholinergic neurons in fear memory, we examined *Hrh1$^{fl/fl}$*, *ChAT-Cre*, and *ChAT-Cre;Hrh1$^{fl/fl}$* mice under recent and remote fear conditioning (Fig. 1a). Compared with control mice, *ChAT-Cre;Hrh1$^{fl/fl}$* male mice displayed similar freezing levels across trials during the conditioning session, suggesting the normal fear learning (Fig. 1b, e). However, the freezing level of *ChAT-Cre;Hrh1$^{fl/fl}$* male mice was increased in the contextual retrieval test but not in the cued retrieval test. Similar results were observed in both recent and remote fear memory (Fig. 1c, d, f, g). We also examined the behavioral features in female *ChAT-Cre;Hrh1$^{fl/fl}$* mice. Female *ChAT-Cre;Hrh1$^{fl/fl}$* mice displayed enhanced contextual fear memory with normal fear learning and cued fear memory, which are comparable to the male *ChAT-Cre;Hrh1$^{fl/fl}$* mice (Fig. 1b–g). To exclude the effect of testing order on the specificity of elevated context fear expression, we conducted experiments that cued fear memory was tested first, followed by contextual fear memory test four hours later (Fig. 1h). Consistent with the behavior in the opposite testing order, *ChAT-Cre;Hrh1$^{fl/fl}$* mice still exhibited enhanced contextual fear memory while displaying normal fear learning and cued fear memory (Fig. 1i–n). These findings showed that the specificity of elevated context fear expression in *ChAT-Cre;Hrh1$^{fl/fl}$* mice is not a consequence of testing order effects.

### $H_1R$ in MS but not other cholinergic brain regions is critical for contextual fear memory

To clarify which population of cholinergic neurons is responsible for the contextual fear memory, we investigated the level of *Hrh1* mRNA in cholinergic neurons in brain samples of wild-type mice after conditioning using RNAscope in situ hybridization (ISH). Interestingly, we observed that *Hrh1* mRNA expression in the medial septum (MS) cholinergic neurons significantly decreased at 1 day, 7 days, 14 days and 28 days post-conditioning compared to controls (Ctrl) (Fig. 2a, b). In contrast, we observed no significant changes in other cholinergic brain regions (Supplementary Fig. 1). In addition, the level of *Hrh1* mRNA in MS GABAergic and glutamatergic neurons was also investigated after conditioning. We found that $H_1R$ expression selectively

decreased in MS cholinergic neurons rather than GABAergic or glutamatergic neurons at different time after conditioning (1, 7, 14, 28 days) (Fig. 2a, b). Consistently, the rheobase of MS cholinergic neurons significantly increased after conditioning at 7–28 days (Fig. 2c, d). Further, we injected a Cre-dependent adeno-associated virus (AAV) containing floxed Hrh1-GFP (AAV-FLEX-Hrh1-GFP, AAV/Hrh1) vector into the MS or the nucleus of the horizontal limb of the diagonal band (HDB)[20,21] of *ChAT-Cre;Hrh1$^{fl/fl}$* mice to specifically restore the expression of $H_1R$ in the MS or HDB cholinergic neurons. The results showed that 97.62% in the MS and 100.00% in the HDB of GFP-labeled neurons were positively stained for choline acetyltransferase (ChAT), and 62.36% in the MS and 55.37% in the HDB of ChAT-positive neurons were infected (Fig. 2e and Supplementary Fig. 2a). Consistent with the behavioral findings in Fig. 1, *ChAT-Cre;Hrh1$^{fl/fl}$* mice injected with AAV/GFP exhibited enhanced contextual fear memory. However, the re-expression of $H_1R$ in the MS area completely rescued their enhanced contextual fear memory, but not cued fear memory (Fig. 2f–h). In contrast, $H_1R$ restoration in the HDB cholinergic neurons failed to alter the contextual or cued fear memory in *ChAT-Cre;Hrh1$^{fl/fl}$* mice (Supplementary Fig. 2b–d). We also engineered a Cre-dependent AAV to selectively knockdown $H_1R$ in cholinergic neurons of *ChAT-Cre* mice. Immunohistochemical analysis showed that 97.92% in the MS and 98.15% in the HDB of GFP-labeled neurons were stained positively for ChAT, and 64.15% in the MS and 55.90% in the HDB of ChAT-positive neurons were infected by the AAV virus (Fig. 2i and Supplementary Fig. 2e). The reduced expression of $H_1R$ in MS cholinergic neurons also contributed to contextual fear memory, rather than cued fear memory (Fig. 2j–l). However, the reduced expression of $H_1R$ in HDB cholinergic neurons has no impact on either contextual fear memory or cued fear memory (Supplementary Fig. 2f–h). These results indicate that $H_1R$ in the MS, rather than other cholinergic brain regions, is critical for enhanced contextual fear memory.

To understand the functional changes associated with $H_1R$ deletion, we performed whole-cell patch-clamp recording of MS cholinergic neurons in *ChAT-Cre* and *ChAT-Cre;Hrh1$^{fl/fl}$* mice (Fig. 2m). We found that the average rheobase current in MS cholinergic neurons from mutant mice was increased, and the spike numbers were significantly lower than that in controls with the increase of the injected currents (Fig. 2n–o). To confirm functional $H_1R$ expression in MS cholinergic neurons, we compared the excitability of MS cholinergic neurons bathed in either ACSF or $H_1R$ antagonist Mepyramine, and found that MS cholinergic neurons showed lower response to the $H_1R$ antagonist Mepyramine (Supplementary Fig. 3a–c). In addition, low *Hrh1* mRNA following knockdown showed low intrinsic excitability, and the response of MS cholinergic neurons to the $H_1R$ antagonist Mepyramine was further reduced (Supplementary Fig. 3d–f), suggesting cholinergic neurons with lower $H_1R$ expression exhibited less sensitive to the $H_1R$ antagonist. These results indicate that the deletion of $H_1R$ in cholinergic neurons reduces the excitability of MS cholinergic neurons, potentially contributing to the observed enhanced contextual fear memory in *ChAT-Cre;Hrh1$^{fl/fl}$* mice.

### The DG is the major downstream region of MS cholinergic neurons responsible for enhanced contextual fear memory in *ChAT-Cre;Hrh1$^{fl/fl}$* mice

To investigate the circuit mechanisms mediating enhanced contextual fear memory in *ChAT-Cre;Hrh1$^{fl/fl}$* mice, we conducted a cleared whole-brain c-Fos mapping procedure to examine the key brain region differentially modulated by the MS. Previous studies suggest that neurons in the dorsal hippocampus and medial prefrontal cortex (mPFC) play a pivotal role in the contextual fear conditioning, and the amygdala has been proposed as the brain region required for cued fear conditioning and also the expression of contextual fear conditioning[22–26]. We quantified the activity-regulated gene c-Fos positive neurons in the subregions of amygdala, dorsal hippocampus

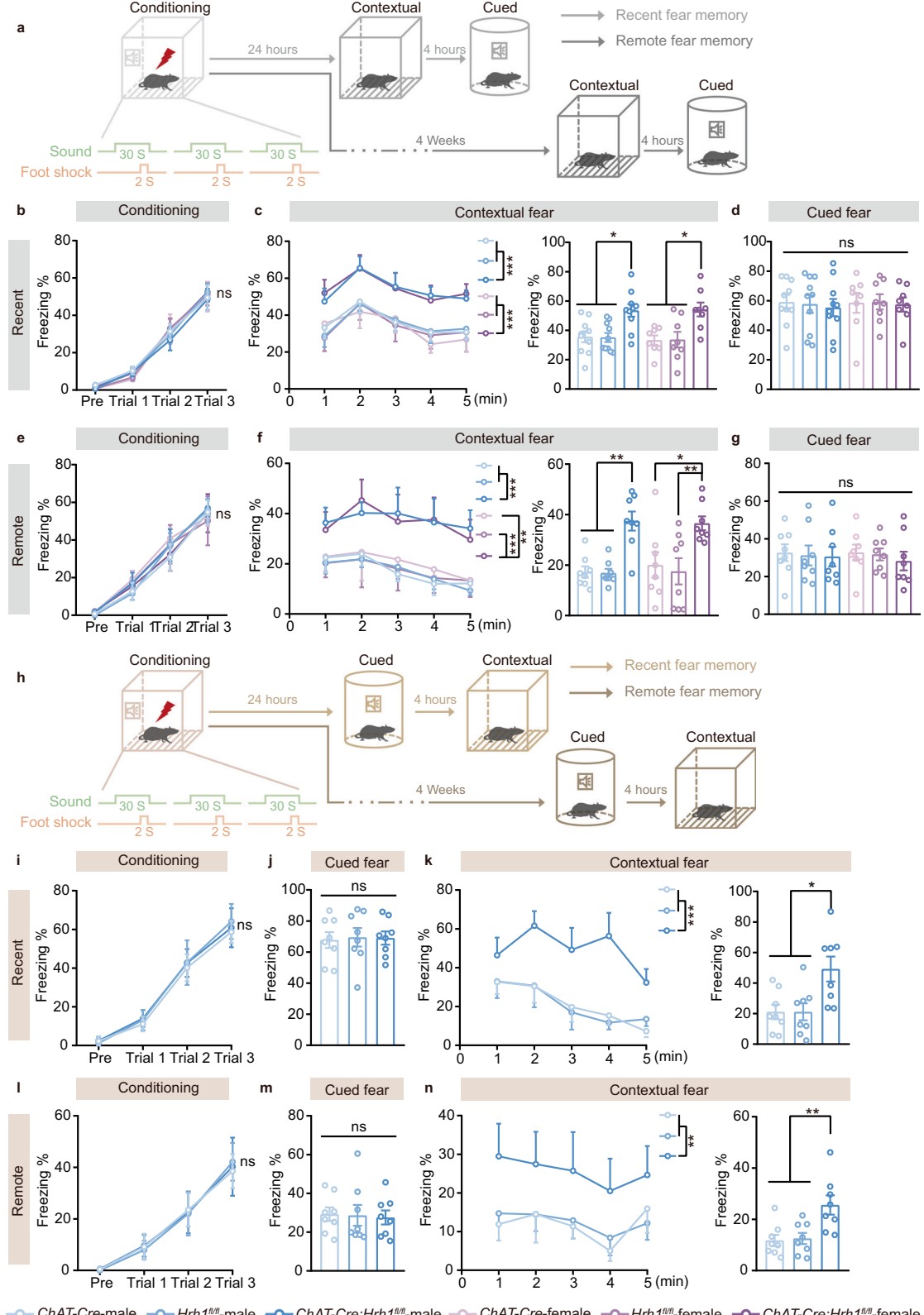

**Fig. 1 | Decreased histamine H₁R expression in cholinergic neurons selectively enhances contextual fear memory. a**, **h** Schematic diagram of the experimental design of recent and remote fear conditioning test. **b**, **e**, **i**, **l** The curve of freezing level during the exploration period and each trial on the conditioning day. **c**, **f**, **k**, **n** The curve of freezing level in each minute (left panel) and the percentage of freezing time during the contextual fear memory retrieval (right panel). **d**, **g**, **j**, **m** The percentage of freezing time during the cued fear memory retrieval. All data are presented as mean ± SEM. *$P < 0.05$, **$P < 0.01$, ***$P < 0.001$, ns, non-significant. Further statistical information and source data are provided as a Source Data file.

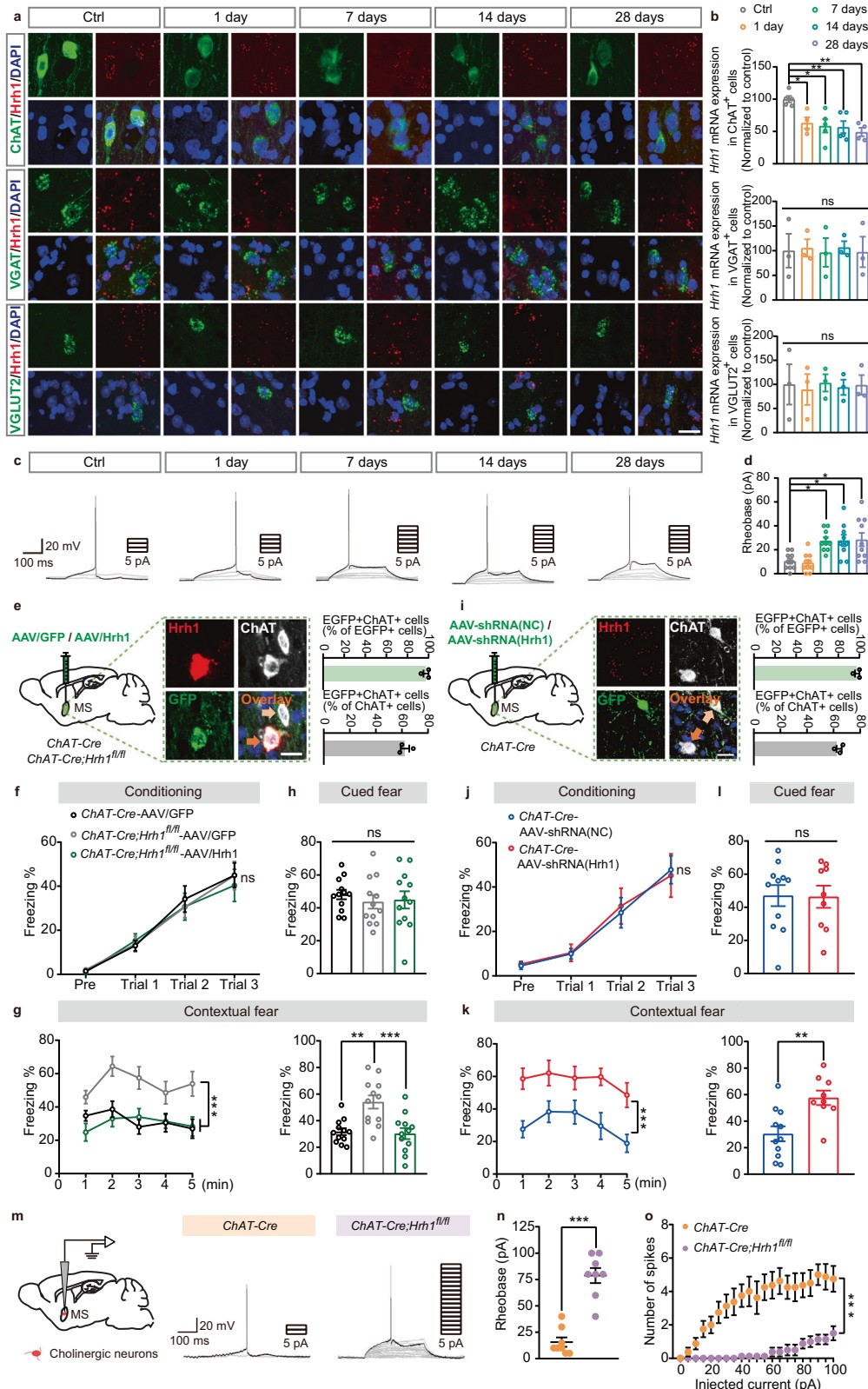

and mPFC to evaluate the response to contextual fear test (Fig. 3a–e). No statistical differences were observed in the number of activated neurons in the basolateral amygdala nucleus (BLA), central amygdala nucleus (CeA), lateral amygdala nucleus (LA), infralimbic cortex (IL), prelimbic cortex (PrL), CA1 and CA3 between *ChAT-Cre* mice and *ChAT-Cre;Hrh1^{fl/fl}* mice. The dentate gyrus (DG) is one of the main downstream targets of the projection from the MS and has been identified as a neural substrate of cognition and mood-related behavior[27,28]. Notably, we found a significantly increased amount of c-Fos positive neurons in the DG (Fig. 3a, b). We also compared the c-Fos expression in *ChAT-Cre* and *ChAT-Cre;Hrh1^{fl/fl}* mice trained but not re-exposed, and found that there were no significant differences in the c-Fos expression of *ChAT-Cre* and *ChAT-Cre;Hrh1^{fl/fl}* mice (Supplementary Fig. 4a, b). In addition, there were no significant differences in their basic c-Fos

**Fig. 2 | H₁R in MS cholinergic neurons is critical for contextual fear memory.**
**a** Representative images of RNAscope in situ hybridization of *Hrh1* mRNA together with choline acetyltransferase (ChAT), *VGAT* mRNA and *VGLUT2* mRNA in the MS after conditioning. Scale bar = 20 μm. **b** Quantitative analysis of *Hrh1* mRNA expression in ChAT⁺, *VGAT*⁺ and *VGLUT2*⁺ cell. **c, d** Threshold current to elicit action potential with the increase of injected currents in MS cholinergic neurons recorded by whole-cell patch-clamp at 1 day, 7 days, 14 days and 28 days post-conditioning. **e–h** *ChAT-Cre;Hrh1^{fl/fl}* mice were injected with AAV-FLEX-Hrh1-GFP (AAV/Hrh1) or AAV-FLEX-GFP (AAV/GFP) in MS. **e** Representative images of *Hrh1* (red), *GFP* (green) and ChAT (white) expression in the MS of *ChAT-Cre;Hrh1^{fl/fl}* mice after the micro-injection of AAV/Hrh1. The percentage of GFP⁺ cells co-expressing ChAT and percentage of ChAT⁺ cholinergic neurons co-expressing GFP in the MS were quantified. Scale bar = 30 μm. *n* = 3 mice. **i–l** *ChAT-Cre* mice were injected with AAV-DIO-EGFP-shRNA(Hrh1)(AAV-shRNA(Hrh1)) or AAV-DIO-EGFP-shRNA(NC)(AAV-shRNA(NC)) in MS. **i** Representative images of *Hrh1* (red), *GFP* (green) and ChAT (white) expression in the MS of *ChAT-Cre* mice after the microinjection of (AAV-shRNA(Hrh1)). The percentage of GFP⁺ cells co-expressing ChAT and percentage of ChAT⁺ cholinergic neurons co-expressing GFP in the MS were quantified. Scale bar, 30 μm. *n* = 3 mice. **f, j** The curve of freezing level during the exploration period and each trial on the conditioning day. **g, k** The curve of freezing level in each minute (left panel) and the percentage of freezing time during the contextual fear memory retrieval (right panel). **h, l** The percentage of freezing time during the cued fear memory retrieval. **m, n** Threshold current to elicit action potential with the increase of injected currents in MS cholinergic neurons recorded by whole-cell patch-clamp. **o** Spike numbers with the increase of injected currents in MS cholinergic neurons recorded by whole-cell patch-clamp. All data are presented as mean ± SEM. *P < 0.05, **P < 0.01, ***P < 0.001, ns, nonsignificant. Further statistical information and source data are provided as a Source Data file.

expression between *ChAT-Cre* mice and *ChAT-Cre;Hrh1^{fl/fl}* mice (Supplementary Fig. 5a, b), suggesting the increased number of c-Fos positive neurons in the DG was not due to the conditional knockout of H₁R in cholinergic neurons. Statistical results indicate that DG was specifically recruited by the retrieval of contextual fear memory (Fig. 3c). These results suggest that the activity of DG neurons may be responsible for the regulation of contextual fear memory in *ChAT-Cre;Hrh1^{fl/fl}* mice.

To examine whether differentially activated neurons in DG were targeted by the MS cholinergic input, we combined an AAV expressing CreERT2 driven by an activity-dependent promoter E-SARE (enhanced synaptic activity-responsive element) and a rabies virus (RABV)-based retrograde monosynaptic tracing system to identify the anatomical connection between the MS cholinergic neurons and retrieval-induced neurons in DG (Fig. 3f, g). Retrograde monosynaptic modified rabies virus tracing showed that the MS cholinergic neurons (53.3% of RV cells were ChAT positive cells) sent ascending monosynaptic inputs to retrieval-induced neurons in DG (Fig. 3h, i), suggesting that retrieval-induced neurons mediate a structural connection from the MS to the DG. Further, we observed the density of c-Fos positive neurons in the DG was significantly increased after cued fear memory (Fig. 3j). Using this functional labeling system (Fig. 3k), we found that the MS cholinergic neurons (11.1% of RV cells were ChAT positive cells) sent ascending monosynaptic inputs to cued fear retrieval-induced neurons in DG, which was much less than the contextual fear retrieval group (53.3% of RV cells were ChAT positive cells) (Fig. 3l, m). Our results suggested the MS-DG cholinergic neural circuitry is highly engaged during contextual fear memory.

### The *ChAT-Cre;Hrh1^{fl/fl}* mice exhibit functional acetylcholine deficiency to DG neurons during the retrieval, but not con-solidation of contextual fear memory
To understand how the functional connectivity between cholinergic neurons and DG neurons dynamically changes and in which phase the alteration occurs, we used the indicators to monitor the acetylcholine (ACh) release in the DG. Because we found that decreased H₁R in cholinergic neurons had no effect on conditioning, the ACh release was recorded 30 min after conditioning (the consolidation phase) or during contextual fear memory test (the retrieval phase). Briefly, an AAV expressing acetylcholine indicators ACh3.0 was injected into the DG of *ChAT-Cre* and *ChAT-Cre;Hrh1^{fl/fl}* mice and optic fibers were installed above the DG for in vivo photometry recordings during the consolidation or retrieval phase of recent contextual fear memory (Fig. 4a, b). Recorded *ChAT-Cre;Hrh1^{fl/fl}* mice exhibited normal fear learning and enhanced contextual fear memory compared with *ChAT-Cre* mice (Fig. 4c, d). By aligning the ACh signals with the video-annotated behavioral epochs in the retrieval phase of recent contextual fear memory, we observed behavior-related changes of ACh release across freezing or mobility bouts (Fig. 4e). During the retrieval phase of recent contextual fear memory, robust decreases in the ACh release usually

occurred in the *ChAT-Cre* mice while slight ACh transients could be elicited in the *ChAT-Cre;Hrh1^{fl/fl}* mice before the onset of freezing bouts (Fig. 4f–h). Data analysis revealed that decreased H₁R expression in cholinergic neurons significantly reduced bout peaks of ACh release (Fig. 4i). Before the onset of mobility bouts, the ACh release in DG of *ChAT-Cre* mice obviously increased while the ACh release in DG of *ChAT-Cre;Hrh1^{fl/fl}* mice was decreased compared to that of *ChAT-Cre* mice (Fig. 4j–l). The overall ACh release in DG of *ChAT-Cre* mice was stronger than that in the *ChAT-Cre;Hrh1^{fl/fl}* mice (Fig. 4m). However, during the consolidation phase of contextual fear memory, freezing and mobility bouts did not elicit such increase or decrease of ACh release in both *ChAT-Cre* mice and *ChAT-Cre;Hrh1^{fl/fl}* mice (Supplementary Fig. 6). These results reveal that the functional acetylcholine deficiency to DG neurons of *ChAT-Cre;Hrh1^{fl/fl}* mice occurs during the retrieval phase instead of the consolidation phase. We next proved that whether MS^{ACh}-DG are functionally connected during the recent contextual fear retrieval phase, AAV-Ef1α-DIO-axon-GCaMP6s was expressed in MS of *ChAT-Cre* and *ChAT-Cre;Hrh1^{fl/fl}* mice, and optical cannula were implanted in DG for fiber photometry (Fig. 4n–p). The fluorescent signal of MS^{ACh}-DG axons in *ChAT-Cre* mice immediately decreased before the onset of freezing bouts and increased before the onset of mobility bouts, while *ChAT-Cre;Hrh1^{fl/fl}* mice showed a flat trace, consistent with the increase or decrease of ACh release during the contextual fear retrieval phase (Fig. 4q–y). The results illustrated the functional connection between MS^{ACh}-DG modulated by H₁R during the contextual fear retrieval phase.

### H₁R in DG-projecting cholinergic neurons in the MS is essential for the retrieval of contextual fear memory
To verify that retrieval-induced neurons in DG do participate in the retrieval of contextual fear memory, we first used AAV-ESARE-ERT2CreERT2 and chemogenetic approach to manipulate retrieval-induced neurons in DG. Wild-type mice were administered tamoxifen before the contextual fear memory test to induce the expression of DREADD (designer receptors exclusively activated by designer drugs) receptor in activated neurons (Supplementary Fig. 7a, b). One week later, hM3Dq activation of retrieval-induced neurons in DG significantly enhanced contextual fear memory (Supplementary Fig. 7c, d). Conversely, the inactivation experiment revealed that the animals treated with clozapine N-oxide (CNO, 1 mg/kg) showed reduced freezing (Supplementary Fig. 7e, f). These results demonstrate that activated DG neurons during contextual fear retrieval play vital roles in the expression of contextual fear memory. Then, we further investigated whether retrieval-induced neurons in DG were involved in the MS-DG circuit regulating the retrieval of contextual fear memory. For optogenetic activation of retrieval-induced neurons in the MS-DG circuit, a viral cocktail containing AAV-ESARE-ERT2-CreERT2, AAV-EF1α-DIO EGFP-TVA-RVG and RV-EnvA-ΔG-ChR2-DsRed was stereotactically injected into the DG of wild-type mice with an optical fiber implanted above the MS (Supplementary Fig. 8a, b). Immunohistochemistry assays confirmed that 61.9% neurons RV cells

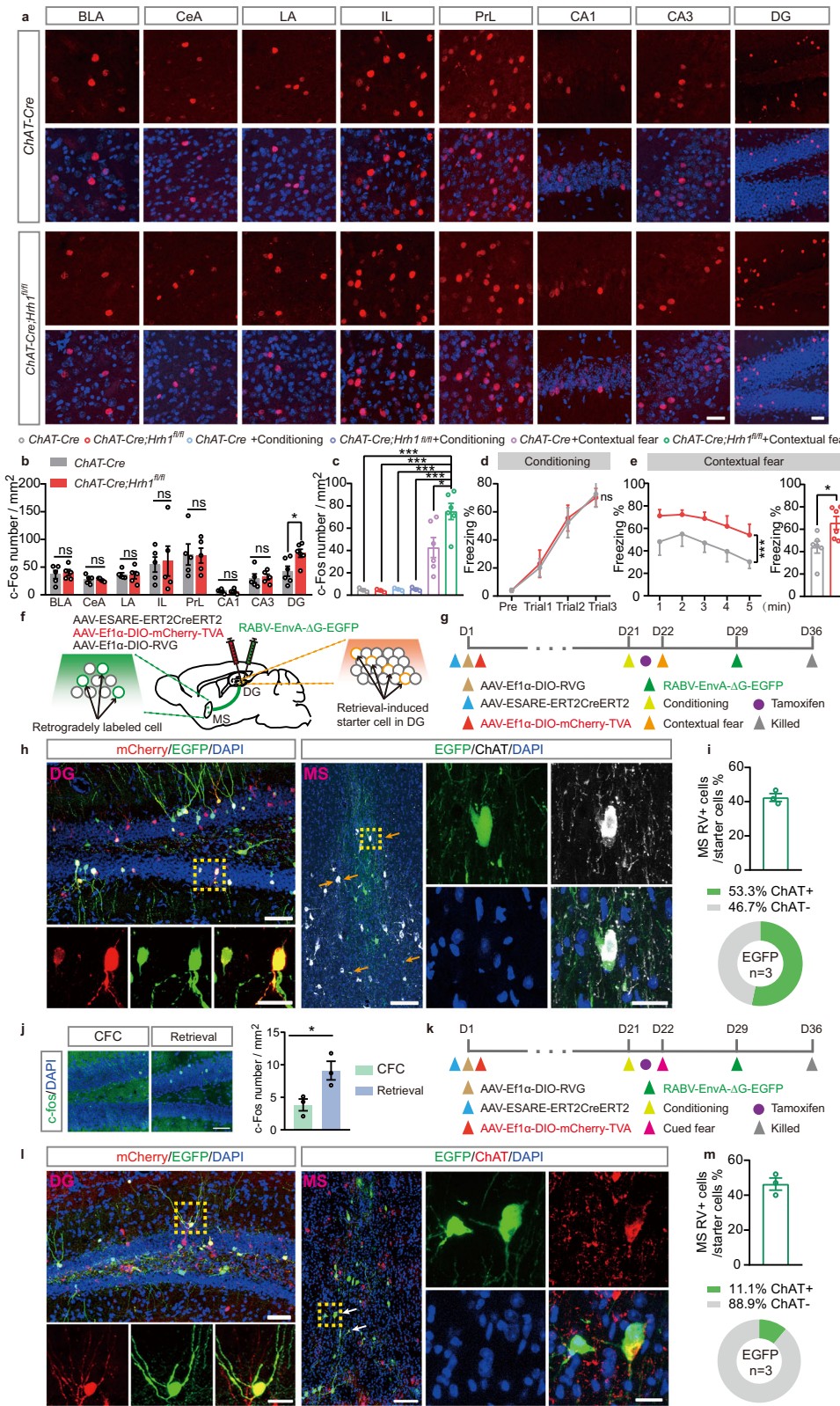

were ChAT positive cells (Supplementary Fig. 8c, d). The optogenetic experiments showed that stimulation of MS neurons during contextual fear retrieval was sufficient to enhance the freezing time upon re-exposure to the same context compared to no stimulation groups (Supplementary Fig. 8e), suggesting the functional connection from the MS to the retrieval-induced neurons in DG. Together, these results indicate that a retrieval-induced neurons in DG are downstream

effectors of MS cholinergic neurons and engage the MS-DG circuit regulating the retrieval of contextual fear memory.

To further address the role of $H_1R$ in MS cholinergic neurons in contextual fear memory retrieval, AAV-DIO-hM3D(Gq)-mCherry was injected into the MS of *ChAT-Cre* and *ChAT-Cre;Hrh1$^{fl/fl}$* mice and CNO (3 μM, 100 nL) was microinjected in the DG 30 min before memory retrieval to intervene in retrieval phase. Immunohistochemical staining

**Fig. 3 | The DG is the major downstream region of MS cholinergic neurons responsible for enhanced contextual fear memory in *ChAT-Cre;Hrh1*$^{fl/fl}$ mice.** **a** Representative images of c-Fos expressing neurons in the amygdala, medial prefrontal cortex and hippocampus of *ChAT-Cre* and *ChAT-Cre;Hrh1*$^{fl/fl}$ mice at 1.5 h after contextual fear memory retrieval. Scale bar = 30 μm. **b** The percentage of c-Fos expressing neurons of the indicated areas. **c** The percentage of c-Fos expressing neurons in DG region of *ChAT-Cre* and *ChAT-Cre;Hrh1*$^{fl/fl}$ mice re-exposed or not. **d** The curve of freezing level during the exploration period and each trial on the conditioning day for (**a**) and (**b**). **e** The curve of freezing level in each minute (left panel) and the percentage of freezing time during the contextual fear memory retrieval (right panel) for (**a**) and (**b**). **f** The schematic for the strategy to retrogradely label MS cholinergic neurons that project to retrieval-induced neurons in DG. **g, k** The timeline of retrograde transsynaptic tracing experiment. **h, l** Left panel: Representative images of EGFP and mCherry doublelabeled starter cells in the DG. Upper panel: scale bar = 50 μm. Lower panel: scale bar = 20 μm. Right panel: Representative images show ChAT (the marker of cholinergic neurons) and EGFP (expressed by retrogradely labeled cells) expression in the MS. Left panel: scale bar = 100 μm. Right panel: scale bar = 20 μm. Arrows indicate EGFP and ChAT doublelabeled cells. **i, m** Upper panel: Quantification of rabies-labeled MS neurons. Lower panel: Percentage of rabies-labeled, immunochemically identified MS cholinergic neurons. *n* = 3 mice per group. **j** Left panel: Representative images of c-Fos expressing neurons in the DG at 1.5 h after cued fear memory retrieval. Scale bar = 50 μm. Right panel: The percentage of c-Fos expressing neurons of the indicated areas. All data are presented as mean ± SEM. *$P < 0.05$, **$P < 0.01$, ns, nonsignificant. Further statistical information and source data are provided as a Source Data file.

showed 97.61% of mCherry-labeled neurons were ChAT$^+$ neurons and 89.64% of ChAT$^+$-labeled neurons were mCherry$^+$ neurons in the MS, indicating that hM3Dq was primarily restricted to the MS cholinergic neurons (Fig. 5a, b). Consistently, intracranial microinjection of CNO completely reversed the enhanced contextual fear memory of the *ChAT-Cre;Hrh1*$^{fl/fl}$ mice to normal level during the retrieval phase (Fig. 5c–e). However, intracranial microinjection of CNO immediately after conditioning has no effect on the behaviors of the *ChAT-Cre;Hrh1*$^{fl/fl}$ mice (Supplementary Fig. 9). These results collectively suggest that H$_1$R in MS cholinergic neurons is critical for the retrieval, but not the consolidation of contextual fear memory. We further demonstrate the precise contribution of H$_1$R in the MS-DG circuit to the contextual fear retrieval by using the Cre-dependent virus for selective knockdown of H$_1$R in the MS cholinergic neurons and the chemogenetic approach to inhibit retrieval-induced neurons in DG of *ChAT-Cre* mice (Fig. 5f, g). CNO (1 mg/kg) was administered by intraperitoneal injection 30 min before memory retrieval. We found that this inhibition produced normal fear learning and cued fear memory, but reversed their enhanced contextual fear memory to normal level (Fig. 5h–j). These data established that selective knockdown of H$_1$R in the MS was sufficient to enhance contextual fear memory by manipulating the retrieval-induced neurons in DG.

We also explored whether the inhibitory muscarinic M$_4$ receptors (M$_4$Rs) were involved in the enhanced recent contextual fear retrieval circuit induced by H$_1$R deficiency in MS cholinergic neurons. The muscarinic M$_4$R antagonist tropicamide, M$_4$R agonist MCN-A-343 or vehicle was bilaterally micro-infused into the DG through cannula 30 min before contextual fear memory retrieval (Supplementary Fig. 10a, b). The vehicle and drug-treated mice showed comparable levels of fear learning and cued fear memory (Supplementary Fig. 10c, e). Importantly, MCN-A-343 administration in the DG effectively reversed enhanced recent contextual fear memory, while tropicamide group exhibited slightly stronger recent contextual fear memory compared to the vehicle group (Supplementary Fig. 10d). These results indicate that M$_4$R in the DG is responsible for the regulation of recent contextual fear memory and is the downstream effector of the enhanced contextual fear retrieval circuit induced by H$_1$R deficiency in MS cholinergic neurons.

## Discussion

In the present study, we found that a deficit of H$_1$R in cholinergic neurons was sufficient to selectively trigger enhanced contextual fear memory, while cued fear memory was normal. We also provide direct evidence that H$_1$R in MS but not other cholinergic brain regions is critical for contextual fear memory. The *ChAT-Cre;Hrh1*$^{fl/fl}$ mice exhibited functional acetylcholine deficiency in DG during the retrieval but not the consolidation of contextual fear memory. Moreover, the changes of contextual fear behavior were causally related to retrieval-induced neurons in DG modulated by H$_1$R in MS cholinergic neurons. Together, these findings underscore the vital role played H$_1$R in cholinergic neurons in contextual fear retrieval.

We found that the *Hrh1* mRNA expression was reduced in the MS cholinergic rather than GABAergic or glutaminergic neurons 1 day, 7 days, 14 days and 28 days after conditioning (Fig. 2a, b). Previous studies have revealed cholinergic neurons in the MS are essential for contextual fear memory generalization and acquisition of fear extinction[29,30]. Pharmacological and optogenetic activation of MS cholinergic neurons enhanced contextual fear conditioning in mice, suggesting these neurons are critical for contextual fear memory. However, MS cholinergic lesions showed no effects on contextual Pavlovian fear conditioning[29]. Despite a large body of the studies, a consensus on the precise functions of MS cholinergic neurons in contextual fear memory remains lacking. Recent studies by using electrophysiology and synaptic tracing methods suggested MS cholinergic subpopulations possess different morphological and physiological properties, and form two structurally defined and functionally distinct subnetworks[31]. Our experiments indicated that selective knockdown of H$_1$R expression in MS cholinergic neurons resulted in the enhancement of contextual fear memory, whereas the enhancement in *ChAT-Cre;Hrh1*$^{fl/fl}$ mice can be reversed by re-expression of H$_1$R in cholinergic neurons in the MS (Fig. 2). Although cholinergic neurons in the HDB are needed for increments in conditioned stimulus (CS) processing, cortical arousal, sustained attentional performance and motor cortex plasticity, and silencing of cholinergic projections from the HDB to the BLA during fear extinction enhances extinction and prevents renewal[30,32], we observed that H$_1$R in cholinergic neurons in the HDB is not involved in the regulation of contextual fear memory (Supplementary Fig. 1 and Supplementary Fig. 2). These results suggested H$_1$R in cholinergic neurons plays distinctive roles at different brain regions and H$_1$R in MS cholinergic neurons may be critical for contextual fear memory. Our recent study demonstrated that deficit of H$_1$R expression in cholinergic neurons resulted in functional deficiency of cholinergic projections from the BF (but not caudate putamen (CPu)) to the prefrontal cortex, and selectively elicits sensorimotor gating ability deficit, social impairments, anhedonia-like behavior and cognitive impairments in several cognition task, including novel object recognition, temporal order memory and Y maze. Our data therefore provide evidence that distinct downstream regions are apparently recruited to modulate cognition impairments arising from different pathological conditions[15]. It is worth noting that the H$_1$R blockade or gene knockout repeatedly produced contradictory results and combined studies in fear conditioning[13,14]. Taking into account the findings of this study and those of our previous study, the actions of H$_1$R depend on the brain region and the specific manipulation is needed. Our findings improve the understanding of the functional role of H$_1$R in brain disorders and provide a selective therapeutic target.

Compared to *ChAT-Cre* mice, the subregion of the hippocampus CA1 and CA3 in *ChAT-Cre;Hrh1*$^{fl/fl}$ mice were unchanged, while temporally specific effect was induced on DG neuronal activation by contextual fear memory retrieval (Fig. 3a–c). These results imply the potential and specific link of DG to H$_1$R in cholinergic neurons. Consistent with the above observations, experimental and computational

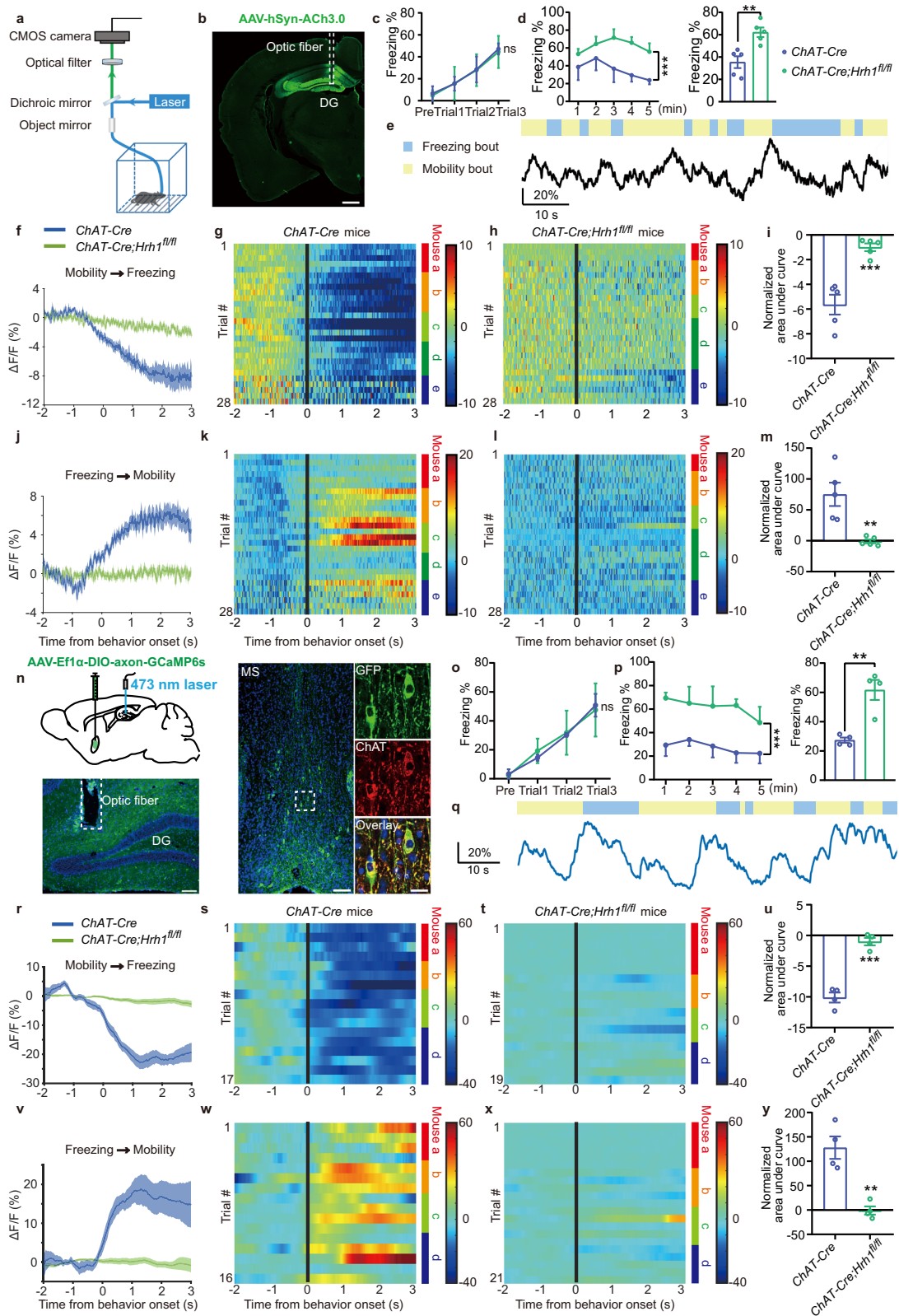

studies have shown the DG is indispensable for processing contextual information and critically involved in the discrimination of similar contexts[33–36]. Indeed, the MS cholinergic input to hippocampus is critical for molecular signaling cascades in the hippocampus that facilitate contextual fear memory formation[37]. The MS cholinergic neurons serve as the predominant source of cholinergic projections to hippocampus[38]. Research measuring ACh levels in the dorsal

hippocampus, pharmacologically manipulating cholinergic receptors in the dorsal hippocampus, or optogenetic activation of MS cholinergic neurons suggest that MS-hippocampus cholinergic circuit is critical for mediating contextual fear conditioning[29]. Our retrograde tracing results indicate a monosynaptic connection between the retrieval-induced neurons in the DG and the cholinergic neurons in the MS (Fig. 3f–i). Further, we found that the chemogenetic inhibition of the

**Fig. 4 | The *ChAT-Cre;Hrh1^{fl/fl}* mice exhibit functional acetylcholine deficiency of MS-DG circuit during the retrieval. a** Schematic illustrating the fiber photometry setup for recording ACh release during the contextual fear retrieval. **b** Illustration of the viral injection site, viral expression, and optic fiber placement (indicated by the white dotted line) in the DG. Scale bar = 500 μm. *n* = 5 mice per group. **c**, **o** The curve of freezing level during the exploration period and each trial on the conditioning day for (**f**–**m**) and (**r**–**y**). **d**, **p** The curve of freezing level in each minute (left panel) and the percentage of freezing time during the contextual fear memory retrieval (right panel) for (**f**–**m**) and (**r**–**y**). **e**, **q** Example trace of ACh release (**e**) and the fluorescence signal of MS projecting axons in DG (**q**) and behavioral epochs during the contextual fear retrieval. Blue boxes above the traces indicate freezing bouts. Yellow boxes above the traces indicate mobility bouts. **f**, **j**, **r**, **v** Averaged plots of ACh release (**f**, **j**) and fluorescence signal of MS projecting axons in DG (**r**, **v**)

aligned to the behavior onset. **g**, **h**, **k**, **l** Heatmap representations of ACh release aligned to the onset of freezing bouts of *ChAT-Cre* mice (**g**, **k**) and *ChAT-Cre;Hrh1^{fl/fl}* mice (**h**, **l**). Color bars at the right of each heatmap represent different individual mice. **i**, **m**, **u**, **y** Normalized AUC of DG acetylcholine release and fluorescence signal of MS projecting axons in DG. **n** Representative image of AAV-DIO-axon-GCaMP6s expression in MS and DG. Left and middle panel: scale bar = 100 μm. Right panel: scale bar = 20 μm. *n* = 4 mice per group. **s**, **t**, **w**, **x** Heatmap representations of fluorescence signal of MS projecting axons in DG aligned to the onset of *ChAT-Cre* mice (**s**, **w**) and *ChAT-Cre;Hrh1^{fl/fl}* mice (**t**, **x**). Color bars at the right of each heatmap represent different individual mice. All data are presented as mean ± SEM. ***P* < 0.01, ****P* < 0.001, ns nonsignificant. Further statistical information and source data are provided as a Source Data file.

retrieval-induced neurons in DG can reverse the enhancement of the contextual fear memory induced by selective knockdown of $H_1R$ in the MS cholinergic neurons by using the Cre-dependent knockdown, engram labeling technology and chemogenetic approach to verify the precise action of $H_1R$ in the $MS^{ACh}$-DG circuit (Fig. 5f–j). However, it can be found that $MS^{ACh} \rightarrow DG$ activation showed an opposite effect (Fig. 5) compared to MS → DG activation on fear memory recall (Supplementary Fig. 8), which seems contradictory. Firstly, the MS is recognized to be involved in the regulation of various memory functions, including fear memory[31,39,40]. But it has three different types of neurons, as known as cholinergic, GABAergic, and glutamatergic neurons, and there are many works proving its differential regulatory role in memory functions[41–43]. Therefore, we speculate that the difference in circuit function between $MS^{ACh} \rightarrow DG$ and MS → DG may be determined by the specific function of diverse MS subpopulations. In addition, in our previous study on the functional feeding circuit, we found that the upstream histaminergic neural projection specifically acts on MS glutamatergic (but not GABAergic or cholinergic) neurons and exerts an inhibitory effect on feeding behavior[44]. And $H_1R$ on different neurons of MS may also receive differential effects from upstream histaminergic neural circuits and be responsible for different regulatory functions in fear memory retrieval. Therefore, there are multiple possible mechanisms of the opposite effects of $MS^{ACh} \rightarrow DG$ and MS → DG activation, and further exploration is needed in the future study.

Furthermore, by applying in vivo ACh recordings, we assessed the real-time alteration of ACh release in DG during the consolidation or retrieval of contextual fear memory. During the retrieval session, we observed decreased ACh release in the DG of *ChAT-Cre* mice but not *ChAT-Cre;Hrh1^{fl/fl}* mice before the onset of freezing bouts (Fig. 4f–i). Moreover, the ACh release in the DG of *ChAT-Cre* mice was increased compared to that of *ChAT-Cre;Hrh1^{fl/fl}* mice before the onset of mobility bouts (Fig. 4j–m). The overall ACh release in the DG of *ChAT-Cre* mice was stronger than that in the *ChAT-Cre;Hrh1^{fl/fl}* mice. Interestingly, this increase or decrease was not observed in *ChAT-Cre* mice or *ChAT-Cre;Hrh1^{fl/fl}* mice during the consolidation session (Supplementary Fig. 6), showing it to be specifically related to the retrieval. Of note, ACh release alteration in the DG region during the retrieval session is not triggered by motor changes since the trace of ACh release is flat during the consolidation session, despite the fact that cholinergic neuromodulation is crucial for the motor circuit operation[45–47]. These results indicate that $H_1R$ in MS cholinergic neurons facilitates ACh release during re-exposure to suppress memory expression. Moreover, although there is some inconsistency in the literature, pharmacological interventions with muscarinic agonists or antagonists support the general premise that muscarinic acetylcholine receptor (mAChR) in the hippocampus appears particularly critical in the acquisition of contextual fear learning during the acquisition and/or consolidation of fear learning. Several studies have indicated that the systemic administration of the scopolamine (mAChR antagonist) before or after training can reduce the acquisition and retrieval of conditioned fear. However, some other studies failed to see effects of

scopolamine or dicyclomine ($M_1$ mAChR antagonist) given after the training session on contextual fear responses even with high doses, suggesting the role of mAChR in fear memory progress is critical and intricate[29]. Accumulating evidence suggests that selective $M_4$ mAChR may offer a novel target for the treatment of psychosis related to cognitive impairments. VU0152100 (a highly selective $M_4$ positive allosteric modulator) blocks amphetamine-induced disruption of the acquisition of contextual fear conditioning[48] and retrieval of remotely acquired contextual memories requires retrosplenial cortex $M_4$ mAChR activity[49]. Further, study using a combination of optogenetic techniques, in vivo and in vitro electrophysiology and multiphoton imaging showed that ACh release from cholinergic septo-hippocampal projections can cause slow inhibition of hippocampal dentate granule cells via astrocyte intermediaries[50]. By regulating granule cell excitability and the size of DG memory ensembles, the cholinergic input from the MS to the hilar perforant path-associated (HIPP) cells in DG control background context fear[28]. Our study indicate that the inhibitory muscarinic $M_4R$ in the DG is the downstream effector of the enhanced contextual fear retrieval circuit induced by $H_1R$ deficiency in MS cholinergic neurons, which further highlights the important role of $M_4R$ in contextual fear memory. We also employed a pathway-specific chemogenetic stimulation protocol by injecting AAV-DIO-hM3Dq-mCherry into the MS region of *ChAT-Cre;Hrh1^{fl/fl}* mice. We found that chemogenetic stimulation of $DG^{ACh}$ terminals during retrieval but not consolidation session led to decreased contextual fear memory compared with their control group (Fig. 5a–e and Supplementary Fig. 9). Our results indicate that $H_1R$ in MS cholinergic neurons can directly induce onset of freezing or mobility bouts by modulating the $M_4R$ in DG during contextual fear memory retrieval.

Histaminergic system in the brain plays a vital role in many pathophysiological processes, especially learning and memory, and has been proposed to be involved in the neuropathology of fear- and emotion-related disorders. In addition, histamine release has been proposed to be a sensitive indicator of stress. For example, histamine turnover rates in the brain are increased in response to stressful stimulation, including the application of electrical shocks or chronic restraint stress[51,52]. Our recent study have shown that $H_3R$ in the vBF cholinergic neurons is responsible for the regulation of contextual fear memory by histamine, suggesting the significant interaction between histaminergic and cholinergic systems in contextual fear memory[53]. As the important member of histamine receptors, $H_1R$ has been found to modulate the activity of cholinergic septal neurons by regulation of its depolarization[54]. However, the precise role of $H_1R$ in cholinergic neurons is still not been clarified. Here, we demonstrate that deletion of $H_1R$ gene in cholinergic neurons in mice is sufficient to enhance contextual freezing during both recent and remote memory test (Fig. 1), resulting from the functional deficiency of cholinergic projections from the MS to DG. The $MS^{ACh} \rightarrow DG$ modulated by $H_1R$ may thus serve as nodal points for time-independent memory retrieval. In addition, the decreased expression of $H_1R$ will weaken the regulation of MS cholinergic neural excitability by histaminergic input during the contextual fear retrieval. In

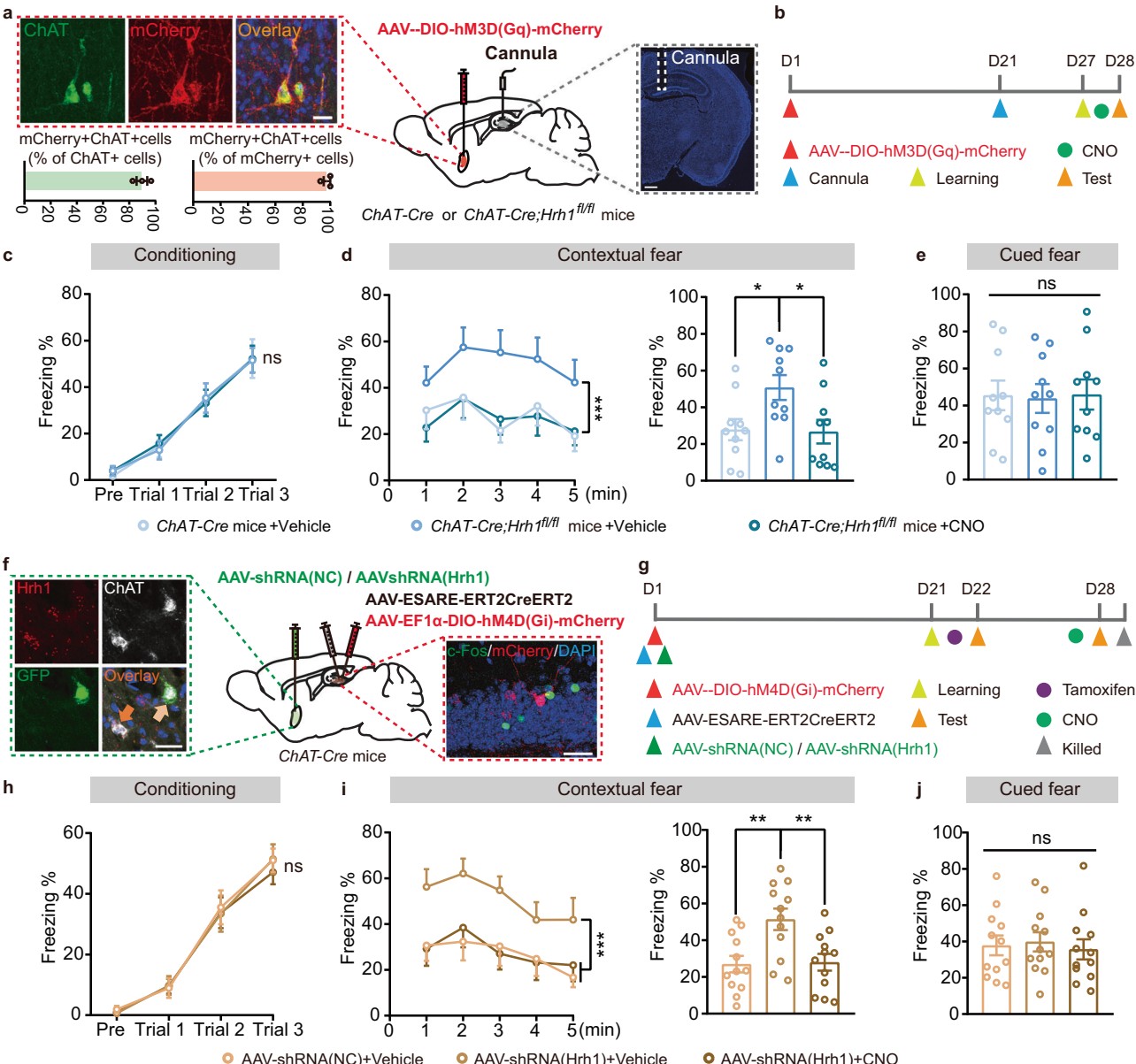

**Fig. 5 | H₁R in DG-projecting cholinergic neurons in the MS is essential for the retrieval of contextual fear memory. a** Left panel: representative images of the hM3Dq expression in the MS. The percentage of mCherry⁺ cells co-expressing ChAT and percentage of ChAT⁺ cholinergic neurons co-expressing mCherry in the MS were quantified. Scale bar = 20 µm. *n* = 3 mice per group. Right panel: representative image of cannula placement (indicated by the white dotted line) in the DG. Scale bar = 500 µm. **b** The timeline of chemogenetic activation experiment. **c**, **h** The curve of freezing level during the exploration period and each trial on the conditioning day. **d**, **i** The curve of freezing level in each minute (left panel) and the

percentage of freezing time during the contextual fear memory retrieval (right panel). **e**, **j** The percentage of freezing time during the cued fear memory retrieval. **f** Left panel: representative images of the AAV-shRNA(Hrh1) expression in the MS. Scale bar = 30 µm. Right panel: representative images show the absence of colocalization between c-Fos and mCherry in DG after CNO administration. Scale bar = 30 µm. **g** Experimental scheme for manipulation of H₁R in MS-DG circuit. All data are presented as mean ± SEM. *$P$ < 0.05, **$P$ < 0.01, ***$P$ < 0.001, ns, non-significant. Further statistical information and source data are provided as a Source Data file.

---

this condition, we found that although H₁R expression on MS cholinergic neurons significantly decreased after 1 day of conditioning session (Fig. 2a, b), their rheobase did not show statistical differences (Fig. 2c, d), indicating other inputs may also affect the excitability of MS cholinergic neurons and be involved in the regulation of contextual fear retrieval. Of note, our results showed that knocking down H₁R in MS cholinergic neurons results in the enhancement of contextual fear retrieval, at least suggesting the involvement of histaminergic system. Thus, further studies are needed to investigate the role of other neural inputs in the contextual fear retrieval.

In conclusion, our results demonstrate that H₁R in the MS cholinergic neurons is crucial for the retrieval of contextual fear memory,

providing a deeper understanding for the molecular and circuit mechanisms underlying fear memory. This finding may pave the way for a better comprehension of vulnerability to PTSD and other pathological fear-related disorders, and highlight the fact that the brain region-specific H₁R could be viewed as an important therapeutic strategy for neurological diseases.

## Methods

### Animals
*Hrh1^{fl/fl}* mice were genetically engineered by standard homologous recombination at the Nanjing Biomedical Research Institute of Nanjing University (Nanjing, China). The exon 3, encoding the core region of

*Hrh1*, was flanked by loxP sequences at both sides. To specifically delete *Hrh1* in ChAT-positive cholinergic neurons, *Hrh1*[fl/fl] homozygous mice were mated with *ChAT-Cre* mice (Jax No. 006410). Wild-type mice (RRID: IMSR_JAX:000664, weight 22–26 g) were purchased from SLAC Laboratory Animal Center (Shanghai, China). Male and female mice aged 8 to14 weeks were used. All mice were bred onto a C57BL/6 J genetic background. The environmental conditions in the mouse facility were: 12 h light/dark cycle (light on from 8:00 a.m. to 8:00 p.m.), temperature range of $22 \pm 2\,°C$, relative humidity range of $50\% \pm 10\%$, and free access to food and water. All procedures complied with the standards of the Institutional Animal Care and was approved by the ethical committee of Zhejiang Chinese Medical University (No. 20230424-06).

## Behavior assays
Mice at 8 to 14 weeks of age were used for all behavioral experiments. They were gently handled daily for five consecutive days and habituated in the testing room for at least 30 min in advance. All behavioral tests were performed during the light phase. All behavioral assays were carried out by examiners blinded to the groups.

## Fear conditioning task
For the Pavlovian fear conditioning paradigm, an initial tone stimulus (30 s, 85 dB) was used as conditioned stimulus (CS) and a scrambled foot shock (0.5 mA, 2 s) was used as unconditioned stimulus (US). Fear conditioning task was carried out in a square-shaped conditioning chamber (Panlab Harvard Apparatus, Spain). This test consisted of three phases: conditioning, contextual fear memory and cued fear memory. On day 1 (conditioning), mice were placed into the fear conditioning chamber with a grid floor capable of delivering foot shocks and allowed to explore freely for 2 min. At the 3, 4 and 5 min, mice were exposed to a 30 s tone (5000 Hz, 85 dB) which is paired with a 2 s electric footshock (0.5 mA), respectively. The mice remained in the conditioning chamber for 60 s. To test recent contextual fear memory, the mice were placed in the conditioning chamber for 5 min on day 2. Four hours after contextual fear memory test, the cued test was carried out in the chamber that different from the conditioning chamber. Mice were allowed to explore freely in the chamber for 2 min, followed by 3 min with tone presented (5000 Hz, 85 dB). After 4 weeks, the tests of remote contextual and cued fear memory were performed as the recent. The behavior paradigm with opposite test order between contextual fear memory test and cued fear memory test could be conducted. Freezing criteria is defined as the complete immobilization of the animal except for respiratory movements. The minimum duration for immobilization to be considered freezing is 2 s. All periods of inactivity with duration lower than 2 s will not be taken into account. The data were analyzed automatically using commercial software (FREEZING, Panlab Harvard Apparatus, Spain).

## In situ hybridization by RNAscope
Mice were sacrificed and perfused with saline and 4% paraformaldehyde in PBS (pH 7.4). The harvested brains were quickly dissected and further fixed in 4% paraformaldehyde for another day before consecutive dehydration in 10%, 20%, and 30% sucrose. Frozen brain slices with 14-μm thickness at a similar coronal position were subjected for ISH. RNAscope Multiplex Fluorescent Reagent Kit v2 (Advanced Cell Diagnostics, 323110) was used for checking *Hrh1* (Advanced Cell Diagnostics, 491141), *VGAT* (Advanced Cell Diagnostics, 319191) and *VGLUT2* (Advanced Cell Diagnostics, 319171) expression. For further double labeling that combines ISH and immunofluorescence, the slices were then incubated with antibody against ChAT (Millipore, AB144P, 1:100) and further photographed by Olympus FV3000 confocal microscope. The images were quantified by FIJI (ImageJ-win64) in a blind manner.

## Immunostaining
Mice were sacrificed and perfused with saline and 4% paraformaldehyde in PBS (pH 7.4). The harvested brains were quickly dissected and further fixed in 4% paraformaldehyde for another day before consecutive dehydration in 10%, 20%, and 30% sucrose. Coronal brain sections were obtained at a thickness of 30-μm using a cryostat (CryoStar NX50, Thermo Fisher). Brain slices were washed 3 times, 5 min each, with PBS. The sections were permeabilized with 0.1% Triton X-100 in PBS for 15 min at room temperature. After blocking by 5% normal donkey serum in PBS for 1 h at room temperature, sections were incubated with anti-ChAT antibody (Millipore, AB144P, 1:100), anti-c-Fos antibody (Synaptic System, 226004, 1:500), at 4 °C overnight and then with AlexaFluor-conjugated secondary antibody-Donkey anti-Goat Alexa Fluor 488 (Invitrogen, A32814, 1:400) or Goat Anti-Guinea pig Alexa Fluor 647 (Abcam, ab150187, 1:400) at room temperature for 2 h. Fluoroshield™ with DAPI (Beyotime, P0131) was used as a nuclear stain. The slices were photographed by Olympus FV3000 confocal microscope and further quantified by FIJI (ImageJ-win64) in a blind manner.

## Viral vectors
AAV-FLEX-GFP (AAV/GFP, $4.5 \times 10^{12}$ V.G./mL), AAV-FLEX-Hrh1-GFP (AAV/Hrh1, $9.0 \times 10^{12}$ V.G./mL), AAV-DIO-EGFP-shRNA(NC) (AAV-shRNA(NC), $3.2 \times 10^{13}$ V.G./mL) and AAV-DIO-EGFP-shRNA(Hrh1) (AAV-shRNA(Hrh1), $1.2 \times 10^{13}$ V.G./mL) were purchased from OBio Biotech Co., Ltd (Shanghai, China). AAV-ESARE-ERT2-Cre-ERT2-PEST ($1.5 \times 10^{13}$ V.G./mL), AAV-hSyn-DIO-hM3D(Gq)-mCherry-WPRE ($2.4 \times 10^{12}$ V.G./mL) and AAV-hSyn-DIO-hM4D(Gi)-mCherry-WPRE ($2.0 \times 10^{13}$ V.G./mL) were purchased from Taitool Bioscience Co., Ltd (Shanghai, China). AAV-EF1α-DIO-mCherry-F2A-TVA-WPRE ($6.0 \times 10^{12}$ V.G./mL), AAV-EF1α-DIO-RVG-WPRE ($5.2 \times 10^{12}$ V.G./mL), RV-ENVA-ΔG-EGFP ($2.0 \times 10^{8}$ V.G./mL), AAV-EF1α-DIO-EGFP-TVA-T2A-RVG-WPRE ($\geq 5.0 \times 10^{12}$ V.G./mL), RV-EnVA-ΔG-ChR2-DsRed ($2.0 \times 10^{8}$ V.G./mL), AAV-hSyn-ACh3.0 ($6.1 \times 10^{12}$ V.G./mL) and AAV-Ef1α-DIO-axon-GCaMP6s ($5.41 \times 10^{12}$ V.G./mL) were purchased from BrainVTA Co., Ltd (Wuhan, China). All viral vectors were aliquoted and stored at −80 °C until use.

## Stereotaxic injections and Optic fiber/Cannula implantation
Animals were anesthetized by intraperitoneal injection of sodium pentobarbital (50 mg/kg) and mounted in a stereotaxic apparatus (RWD Life Science, Shenzhen, China). The body temperature of anesthetized mice was maintained at 37 °C using a heating pad during the total operation. A total of 300 nL of virus was stereotaxically injected through a glass microelectrode controlled by a micropump (Micro 4, World Precision Instruments, USA, WPI) at a rate of 50 nL/min. The needle was left at the injection site for approximately 5 min and then withdrawn slowly. The stereotaxic coordinates were AP + 1.0 mm, ML ± 0.0 mm, and DV −4.0 mm for MS injection, AP + 0.75 mm, ML ± 0.8 mm, and DV −5.0 mm for HDB injection, AP −2.0 mm, ML ± 1.0 mm, and DV −1.8 mm for DG injection. Optic fibers (Inper Ltd., China) and cannula (RWD Life Science, China) were implanted and held in MS (AP + 1.0 mm, ML ± 0.0 mm, DV −4.0 mm) or DG (AP −2.0 mm, ML ± 1.0 mm, DV −1.8 mm) three weeks after virus infusion. After surgery, mice were returned to the home cage and carefully monitored for 48 h. Mice were used in experiments 3–4 weeks after virus injection and 1 week after implantation. The sites of optic fiber/cannula implantation or viral expression were histochemically verified at the end of all experiments following transcardiac 4% PFA perfusion and brain sectioning. Mice that showed incorrect sites were excluded from all relevant analyses.

## E-SARE induction experiments
Mice received bilateral microinjection of viral cocktail, and fear conditioning task was performed at three weeks after virus injection. The 20 mg/mL tamoxifen (Sigma-Aldrich, CAS#10540-29-1) was prepared

in corn oil and administered intraperitoneally (i.p.) at a dose of 80 mg/kg before 3–4 h of contextual fear test on the next day.

## Optogenetic and Chemogenetic intervention

For optogenetic intervention, implanted optic fibers were connected to a laser source using an optical fiber sleeve (Inper Ltd., China). Continuous 473 laser stimulation (20 Hz, 10-ms pulse, 5 mW) was applied for MS cholinergic neurons during the whole period of contextual fear memory test. For chemogenetic intervention, CNO was administered either by i.p. injections at 1 mg/kg or by intra-DG injections (3 µM, 100 nL per side) bilaterally at 30 min before the behavior test. The equal volume of saline was administered by either i.p. injection or intra-DG injections for control groups.

## Fiber photometry

The AAV encoding ACh3.0 was injected into DG and AAV-Ef1α-DIO-axon-GCaMP6s was injected into MS of *ChAT-Cre* or *ChAT-Cre;Hrh1^fl/fl* mice. After three weeks of expression, mice were implanted with optic fibers (0.23 mm O.D., 0.37 mm numerical aperture (NA); Inper Ltd., China) above the DG. Fiber photometry was performed one week after implantation surgery. The fiber photometry system (Thinker Tech, Nanjing, China) was used to record the fluorescence signals in freely moving mice. Purple LED light (405/10 nm, model 65133, Edmund Optics) and blue LED light (470/10 nm, model 65144, Edmund Optics) were bandpass filtered, reflected by a dichroic mirror (model 67-069,495 nm long pass, Edmund Optics) and a dichroic mirror (model 87-282, multi-band filter, Edmund Optics), and then focused using a 20× objective lens (Olympus). An optical fiber guided the light between the commutator and the implanted optical fiber cannula. The green fluorescence was bandpass filtered (525/39 nm, model MF525-39, Thorlabs) and collected using a CMOS camera (U3-3260SE Rev.1.2, IDS Imaging). Photometry data were exported as MATLAB Mat files for further analysis. Analysis code is available at GitHub repository: https://github.com/wellsjay/TrippleColorMultiChannelAnalysisPackage.git.

For the acquisition and analysis of all fiber photometry data, the parameter settings are consistent and unified. Time 0 was aligned to the onset of a behavioral epoch. F0 is the baseline average fluorescence signals of the 2 s before Time 0. The fluorescence responses were indicated by ΔF/F (calculated as (F-F0)/F0). The area under the curve of the ΔF/F plot was measured to quantify the response to contextual fear retrieval in the DG. The minimum length of freezing bout considered for fiber photometry is 2 s, which is consistent with the freezing criteria of fear conditioning task.

## Electrophysiology

The mice were quickly perfused with ice-cold artificial cerebrospinal fluid (ACSF) containing in mM: 120 NaCl, 11 Dextrose, 2.5 KCl, 1.28 $MgSO_4$, 3.3 $CaCl_2$, 1 $NaH_2PO_4$, and 14.3 $NaHCO_3$. The brain was quickly transferred to a vibratome (VT1000 mol/L/E, Leica) to prepare coronal slices at 300-µm thickness and immersed in ice-cold ACSF constantly bubbled with 95 % $O_2$ and 5% $CO_2$ for 1 min. Coronal slices containing the MS were recovered in a chamber filled with ACSF saturated with 95% $O_2$ and 5% $CO_2$ at 37 °C for 30 min and incubated at 25 °C for 1 h before recording. Then the slices were transferred into a recording chamber at 25 °C for patch clamp recording. The patch pipette with resistances ranging from 5 to 10 MΩ was filled with recording solution (containing in mM: 5 NaCl, 140 K-gluconate, 0.2 EGTA, 2 Mg-ATP and 10 HEPES). Signals were amplified and recorded by HEKA EPC10 amplifier (HEKA Instruments, Germany). To test the action potential threshold of neurons being recorded, episodic currents were injected under the current clamp configuration in 5 pA increments from 0 pA to depolarizing 100 pA.

## Drugs

Mice were micro-infused with saline, M4R agonist MCN-A-343 (79 mM, MedChemExpress), M4R antagonist tropicamide (56 mM,

MedChemExpress) into two sides of the DG via an implanted cannula 30 min before the test. The total injection value was 500 nL per mouse. In electrophysiology experiments, the drug was applied by bath application. The mepyramine (GLPBIO, GC11291) was freshly prepared in ACSF and equilibrated with 95% $O_2$ and 5% $CO_2$ before perfusing the slices.

## Statistical analysis

Number of experimental replicates (*n*) is indicated in figure legend and refers to the number of experimental subjects independently treated in each experimental condition. No statistical methods were used to pre-determine sample size, or to randomize. All datasets were tested for Gaussian distribution using a Shapiro–Wilk normality test. Two datasets were statistically compared using a Student's *t* test if the null hypothesis of normal distribution was not rejected. ANOVA tests were used when comparing more than two normally distributed datasets. In case of non-normal data distribution, non-parametric tests were used: Mann–Whitney *U* test was used for single comparisons, and the Scheirer Ray Hare test for two-way analysis of variance. Statistical analyses were carried out using Prism (version 8.0) or SPSS (version 25.0). A statistical significance threshold was set at 0.05, and significance levels are presented as $*P \leq 0.05$, $**P \leq 0.01$ or $***P \leq 0.001$ in all figures.

## Reporting summary

Further information on research design is available in the Nature Portfolio Reporting Summary linked to this article.

## Data availability

All data needed to evaluate the conclusions in the paper are presented in the results and/or supplementary materials. Any additional information is available from the corresponding author. Source data are provided with this paper.

## Code availability

Analysis code[55] is available at this GitHub repository: https://github.com/wellsjay/TrippleColorMultiChannelAnalysisPackage (https://zenodo.org/records/11363248).

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

## Acknowledgements

This project was supported by grants from the National Key R&D Program of China (2021ZD0202803 to Z.C.), the National Natural Science Foundation of China (U21A20418 to Z.C., 82104138 to L.C.), the Zhejiang Provincial Natural Science Foundation of China under Grant No. LY24H310003 to L.C. and the China Postdoctoral Science Foundation (BX2021270 to L.C., 2021M692894 to L.C.).

## Author contributions

Z.C., L.C. and W.K.L. designed research; L.C., L.X., M.Z.L., J.Y.L., X.Y.Q. and M.H.L. performed research; L.C., L.X., W.K.L. and M.Z.L. analyzed data; Z.C., W.K.L., Y.R.Z., C.L.X. and Y.W. supervised the experiments. Z.C., L.C., W.K.L. and Y.R.Z. wrote the paper. All authors contributed to the final version of the manuscript.

## Competing interests

The authors declare no competing interests.
