## [Peer Review File · Nature Communications]

Histamine H1 receptors in dentate gyrus-projecting cholinergic neurons of the medial septum suppress contextual fear retrievalREVIEWER COMMENTS

Reviewer #1 (Remarks to the Author):

This is an interesting paper studying the role of the histamine receptor HR1 in cholinergic cells (Chat+) of the medial septum (MS) in fear memory. Mechanisms controlling the expression of previously acquired memories is an important field of research due to its potential clinical implications, whether to repress them in cases of intrusive memories or to enhance them to treat memory deficits.

The authors provide substantial evidence using viral tools, genetic models, fluorescent reporters, and pharmacology that HR1 in cholinergic septal cells facilitates their cellular activation during the retrieval test, leading to acetylcholine (ACh) release in the dentate gyrus and limiting the expression of memory.

While the authors claim in the title that septal H1R mediates fear retrieval, their collective results in fact describe a mechanism that opposes retrieval (i.e. because in the absence of HR1 memory is elevated). Thus, the authors might want to reconsider the title for clarity.

Methods are sufficiently described. It would help to be more concise when describing experimental effects, e.g. to state "it increased / decreased" rather than ambiguous terms such "affected, altered, etc".

Representation of the literature and current knowledge:

The authors should discuss how their results integrate with the shown ability of septal ACh release in the DG to promote granule cell disinhibition, and the importance of septal ACh release in learning and consolidation. Including any previous knowledge of histamine release in the brain during learning vs memory recall would also benefit the work.

Specific comments:

1- Global deletion of H1R from Chat-cre cells spared conditioning and tone fear, but it enhanced recent and remote contextual fear. Such enhancement is linked with downregulation of Hrh1 mRNA expression after conditioning, but its presence cannot be seen in the red channel of panel 2A, including at baseline. It is therefore unclear how Hrh1 was quantified, or whether its quantification was limited to Chat cells or other cell types. The same applies to panel 2G, where knockdown of Hrh1 does not seem to make a difference in the intensity level of Hrh1.

2- Given the above, an important test to confirm functional HR1 expression in Chat cells at baseline (and the effective KD by miRNA) would be to show loss of response to a histamine agonist/HR1 antagonist following KD, perhaps using a similar approach as in panel 2K, or an equivalent cellular readout.

The authors used an elegant AAV strategy to conditionally re-express Hrh1 in Chat-MS cells in a global KO background (Fig.2c-f), and found that such manipulation is sufficient to normalize (i.e. reduce to control level) fear memory. Conversely, conditional KD of Hrh1 with AAV miRNA injected in MS of Chat-cre mice was sufficient to enhance fear memory (Fig.2h-j), similar to global Hrh1 deletion.

From these behavioral data, it can be concluded that HR1 in MS limits fear memory expression.

3- The authors studied the brain regions downstream of MS-Chat that support the effect of

HR1. They examined cFos expression after retrieval, and found a selective increase in DG granule cells of cKO mice.

Without a group of mice trained but not re-exposed, it is not possible to know what regions were specifically recruited by retrieval. It is also desirable to know if increased DG activation was paralleled by reduced cFos of MS-Chat cells in cKO, as implied from their circuit model. Finally, a statistical comparison of main Fig.3 and supplementary Fig.3 would help identify the impact of cKO on behavioral activation.

4-The authors used retrograde tracing with an inducible, activity-dependent viral vector to label MS cells that project to active DG cells during retrieval. They found that about half of MS afferents were Chat(+). Please clarify the statistical test performed here and purpose (panel 3F).

5-Fiber photometry of the acetylcholine reporter Ach3 was used to assess the impact of HR1 deletion in Chat cells on Ach levels in the DG. An inverse relationship between Ach levels and freezing was observed. Ach fluctuations were largely ablated in cKO mice, indicating that HR1 in Chat cells facilitates Ach release during re-exposure to block memory expression.

However, the authors showed that Hrh1 deletion in Chat cells reduces the spike probability in response to depolarization. Because conditioning itself persistently decreased Hrh1 expression in WT, it is not clear then how Ach would be preferentially released during retrieval. It is therefore important to (a) define the cell type undergoing Hrh1 downregulation after training, (b) test if after conditioning the rheobase of MS-Chat cells increases, and (c) discuss this potential conundrum.

6- At the circuit level, their data imply that Ach in DG blocks granule cell activation, and that (excess) DG activation in cKO impairs memory recall. They confirmed that chemogenetically activating MS-Chat terminals in the DG of cKO mice, or inhibiting granule cells themselves, normalizes (i.e. decreases) memory. In contrast, optogenetic activation of all MS cells projecting to DG during the retrieval test increases memory. Such opposite effects of MS-Chat→DG activation and MS-pan→DG activation should be directly discussed (e.g. different HR1 expression, diverse MS populations, etc). Also, please state in the main text remote vs recent memory tests in these last experiments.

In a final experiment, the authors found that an agonist of Gi-coupled M4Rs infused in DG reduced memory in cKO mice, providing a plausible mechanism for the effect of HR1 in DG inhibition and memory.

Reviewer #2 (Remarks to the Author):

Chen and colleagues provide a well-presented manuscript containing a novel line of inquiry that investigates the contribution of the H1 histamine receptor (H1R) within the medial septum (MS) to dentate gyrus (DG) pathway during expression of context fear behavior. The authors use the multiple techniques of transgenic mice, fiber photometry, chemogenetics, ex vivo electrophysiology, optogenetics, neuropharmacology, and intersectional AAV expression to build the case that the absence of H1R in MS cholinergic neurons projecting to the DG enhances, specifically, context fear expression/retrieval. They show that ChAT-Cre;Hrh1 mice show reduced excitability (rheobase) of ACh MS neurons. Furthermore, cholinergic MS projecting to context fear memory engram cells in the DG, and the cholinergic activity of ChAT-Cre;Hrh1, is reduced during context fear expression, as

measured by fiber photometry. Finally, using chemogenetic methods the authors demonstrate that activating the MS-DG pathway with CNO reduces the elevated context fear phenotype in ChAT-Cre;Hrh1 mice. The claims of the authors are highly specific regarding neuronal circuitry and behavior; moreover, investigating the role of histamine receptors in the expression of defensive behavior is a novel and important advance to the field. However, there are several limitations in the current form of the manuscript that reduce this reviewer's enthusiasm. Below are suggestions to be addressed prior to publication.

Major concerns:

a) The behavioral results in Fig 1 are strong and convincing. However, since the effects of H1R on cued and context fear is a novel line of inquiry, the authors should demonstrate that the specificity of elevated context fear expression is not a consequence of testing order effects, i.e., a control study that tests cued fear first then four hours later tests context fear is required.

b) The importance of the MS-DG pathway is a central feature of this manuscript, for this reason understanding the relationship between DG activation and context freezing behavior is essential. The addition of context freezing behavior for control and ChAT-Cre;Hrh1 mice used in Fig 3a-b would further support the specificity of DG activation level (as measured by c-fos in Fig 3) and heightened context freezing behavior.

c) The use of fiber photometry and the Ach3.0 is a strong contribution to this manuscript. However, some questions remain for the recordings and analyses in Fig 4. Panels g and k collapse freezing trials across all mice and statistically compare "area under the curve"; however, the robustness of the effects in panels d-k between mouse subjects would be supported by an analysis that compared area under the curve for individual ChAT-Cre;Hrh1 and ChAT-Cre mice (i.e., take the average dF/F per mouse and compare between ChAT-Cre;Hrh1 and ChAT-Cre groups). Providing the context freezing levels of the mice in this fiber photometry experiment would also help support the relationship between MS ACh in the DG and normal or heightened context fear behavior. Finally, it is unclear if only freezing bouts greater than 3 seconds were considered in this analysis.

d) Interpretation of Fig 5 requires the addition of control groups to assess the elevated context fear in ChAT-Cre;Hrh1 (Fig 5d) and miR30shRNA knockdown (Fig. 5i) by adding the ChAT-Cre group (or a genotype that will express "normal" levels of context fear). This will support the claim that the DREADD manipulations restored context freezing behavior to "normal" levels.

e) The specificity of the MS-DG context vs. cued fear retrieval circuitry would be further supported by the inclusion of a cued fear memory retrieval group that undergoes the retrograde and engram tagging procedure in Fig 3c-f. Because all mice undergo fear conditioning with 3 tone cue & shock pairing, they are all encoding the tone-shock association. However, the specificity of the context vs cued freezing effect depends on the MS-DG circuitry being engaged only when context retrieval is occurring. For this reason, a quantification of cell activated during cued fear retrieval within the MS-DG pathways (as shown in Fig 3f) is necessary.

Minor concerns:

a) Consistent use of "consolidation" period and "conditioning" session would help clarify when fiber photometry (and other manipulations) are being implemented.

- b) Additional details are required in the methods section that specify the product numbers for antibodies used (e.g., cFos, secondary, DAPI, ChAT).
- c) The abstract mentions “reciprocally connected” MS ACh neurons and DG cells, but I did not see an investigation of DG projections to MS ACh neurons. Alternative word choice is recommended.
- d) Freezing was quantified with PANLAB software, but the criterion for freezing behavior is not stated. Please add the freezing criteria.
- e) Line 341 mentions a “behavioral closed-loop” chemogenetic manipulation. The chemogenetics manipulations did not appear to be “closed-loop”; please consider re-wording or expanding on why the use of DREADDs was closed-loop.
- f) Please add labels to the color bars in figure 4.
- g) The discussion could include more information about H1R physiology in the MS and how this receptor is regulated by experiential factors like fear conditioning.
- h) The introduction and discussion should include additional information about the muscarinic receptor to integrate the results in Supp Fig 7 with the rest of the manuscript’s results.
- i) Based on the limited amount of data that explores the 28-day remote time point of context memory fear retrieval, the amount of discussion on systems consolidation results should be tempered.

Point-by-point response to reviewers' comments

REVIEWER COMMENTS

Reviewer #1 (Remarks to the Author):

This is an interesting paper studying the role of the histamine receptor HR1 in cholinergic cells (Chat⁺) of the medial septum (MS) in fear memory. Mechanisms controlling the expression of previously acquired memories is an important field of research due to its potential clinical implications, whether to repress them in cases of intrusive memories or to enhance them to treat memory deficits.

The authors provide substantial evidence using viral tools, genetic models, fluorescent reporters, and pharmacology that HR1 in cholinergic septal cells facilitates their cellular activation during the retrieval test, leading to acetylcholine (ACh) release in the dentate gyrus and limiting the expression of memory.

While the authors claim in the title that septal H1R mediates fear retrieval, their collective results in fact describe a mechanism that opposes retrieval (i.e. because in the absence of HR1 memory is elevated). Thus, the authors might want to reconsider the title for clarity.

Methods are sufficiently described. It would help to be more concise when describing experimental effects, e.g. to state "it increased / decreased" rather than ambiguous terms such "affected, altered, etc".

Representation of the literature and current knowledge:

The authors should discuss how their results integrate with the shown ability of septal ACh release in the DG to promote granule cell disinhibition, and the importance of septal ACh release in learning and consolidation. Including any previous knowledge of histamine release in the brain during learning vs memory recall would also benefit the work.

Response: We thank reviewer for the positive and constructive comments, which have significantly helped us improve the manuscript. We have revised the title of manuscript as "Histamine H₁ receptors in dentate gyrus-projecting cholinergic neurons of the medial septum suppress contextual fear retrieval". We also proofed our manuscript carefully and corrected ambiguous terms. Combined with the minor concerns g) and h) from reviewer #2, we have added comment in the revised discussion as follows:

Lines 319-327 in revised manuscript – Pharmacological and optogenetic activation of MS cholinergic neurons enhanced contextual fear conditioning in mice, suggesting these neurons are critical for contextual fear memory. However, MS cholinergic lesions showed no effects on contextual Pavlovian fear conditioning¹. Despite a large body of the studies, a consensus on the

precise functions of MS cholinergic neurons in contextual fear memory remains lacking. Recent studies by using electrophysiology and synaptic tracing methods suggested MS cholinergic subpopulations possess different morphological and physiological properties, and form two structurally defined and functionally distinct subnetworks².

Lines 360-365 in revised manuscript – The MS Cholinergic neurons serve as the predominant source of cholinergic projections to hippocampus³. Research measuring ACh levels in the dorsal hippocampus, pharmacologically manipulating cholinergic receptors in the dorsal hippocampus, or optogenetic activation of MS cholinergic neurons suggest that MS-hippocampus cholinergic circuit is critical for mediating contextual fear conditioning¹.

Lines 426-450 in revised manuscript – Although there is some inconsistency in the literature, pharmacological interventions with muscarinic agonists or antagonists support the general premise that muscarinic acetylcholine receptor (mAChR) in the hippocampus appears particularly critical in the acquisition of contextual fear learning during the acquisition and/or consolidation of fear learning. Several studies have indicated that the systemic administration of the scopolamine (mAChR antagonist) before or after training can reduce the acquisition and retrieval of conditioned fear. However, some other studies failed to see effects of scopolamine or dicyclomine (M1 mAChR antagonist) given after the training session on contextual fear responses even with high doses, suggesting the role of mAChR in fear memory progress is critical and intricate¹. Accumulating evidence suggests that selective M4 mAChR may offer a novel target for the treatment of psychosis related to cognitive impairments. VU0152100 (a highly selective M4 positive allosteric modulator) blocks amphetamine-induced disruption of the acquisition of contextual fear conditioning⁴ and retrieval of remotely acquired contextual memories requires retrosplenial cortex M4 mAChR activity⁵. Further, study using a combination of optogenetic techniques, in vivo and in vitro electrophysiology and multiphoton imaging showed that acetylcholine release from cholinergic septohippocampal projections can cause slow inhibition of hippocampal dentate granule cells via astrocyte intermediaries⁶. By regulating granule cell excitability and the size of DG memory ensembles, the cholinergic input from the MS to the hilar perforant path-associated (HIPPA) cells in DG control background context fear⁷. Our study indicate that the inhibitory muscarinic M4R in the DG is the downstream effector of the enhanced contextual fear retrieval circuit induced by H1R deficiency in MS cholinergic neurons, which further highlights the important role of M4R in contextual fear memory.

Lines 457-468 in revised manuscript – Histaminergic system in the brain plays a vital role in many pathophysiological processes, especially learning and memory, and has been proposed to be involved in the neuropathology of fear-

and emotion-related disorders. In addition, histamine release has been proposed to be a sensitive indicator of stress^{8,9}. For example, histamine turnover rates in the brain are increased in response to stressful stimulation, including the application of electrical shocks or chronic restraint stress. Our recent study showed that H₃R in the vBF cholinergic neurons is responsible for the regulation of contextual fear memory by histamine, suggesting the significant interaction between histaminergic and cholinergic systems in contextual fear memory¹⁰. As the important member of histamine receptors, H₁R has been found to modulate the activity of cholinergic septal neurons by regulation of its depolarization¹¹. However, the precise role of H₁R in cholinergic neurons is still not been clarified.

Specific comments:

1- Global deletion of H1R from Chat-cre cells spared conditioning and tone fear, but it enhanced recent and remote contextual fear. Such enhancement is linked with downregulation of *Hrh1* mRNA expression after conditioning, but its presence cannot be seen in the red channel of panel 2A, including at baseline. It is therefore unclear how *Hrh1* was quantified, or whether its quantification was limited to Chat cells or other cell types. The same applies to panel 2G, where knockdown of *Hrh1* does not seem to make a difference in the intensity level of *Hrh1*.

Response: We are sorry for the blurry *in situ* hybridization images. We have showed the enlarged images below, and provided the raw figure with AI version in the submission system. For quantification of *Hrh1*, we drew a ROI as the outline of cholinergic, GABAergic or glutaminergic neurons, and calculate the *Hrh1* expression within the outline using the FIJI (ImageJ-win64). The quantification of *Hrh1* therefore was limited to cholinergic, GABAergic or glutaminergic neurons. We have added these data into Fig.2a and manuscript as follows:

Lines 115-120 in revised manuscript – In addition, the level of *Hrh1* mRNA in MS GABAergic and glutamatergic neurons was also investigated after conditioning. We found that H₁R selectively decreased expression in MS cholinergic neurons rather than GABAergic or glutamatergic neurons at different time after fear conditioning (1,7,14,28 days) (Fig. 2a and b).

Figure 2a. Representative images of RNAscope in situ hybridization of *Hrh1* mRNA together with choline acetyltransferase (ChAT), *VGAT* mRNA and *VGLUT2* mRNA in the MS after contextual fear conditioning. Scale bar = 20 μ m.

Figure 2i. Representative images of *Hrh1* (red), GFP (green) and ChAT (white) expression in the MS of *ChAT-Cre* mice after the microinjection of (AAV-miR30shRNA(*Hrh1*)). The percentage of GFP⁺ cells co-expressing ChAT and percentage of ChAT⁺ cholinergic neurons co-expressing GFP in the MS were quantified. Scale bar, 30 μ m. n = 3 mice.

2- Given the above, an important test to confirm functional HR1 expression in Chat cells at baseline (and the effective KD by miRNA) would be to show loss of response to a histamine agonist/HR1 antagonist following KD, perhaps using a similar approach as in panel 2K, or an equivalent cellular readout.

The authors used an elegant AAV strategy to conditionally re-express *Hrh1* in Chat-MS cells in a global KO background (Fig.2c-f), and found that such manipulation is sufficient to normalize (i.e. reduce to control level) fear memory. Conversely, conditional KD of *Hrh1* with AAV miRNA injected in MS of Chat-

cre mice was sufficient to enhance fear memory (Fig.2h-j), similar to global *Hrh1* deletion.

From these behavioral data, it can be concluded that HR1 in MS limits fear memory expression.

Response: Thank you very much for this valuable comment. We compared the excitability of MS cholinergic neurons bathed in either ACSF or H₁R antagonist mepyramine, and found that MS cholinergic neurons showed lower response to the H₁R antagonist (Supplementary Fig. 3a-c). In addition, low *Hrh1* mRNA following knockdown showed low intrinsic excitability, and the response of MS cholinergic neurons to the H₁R antagonist mepyramine was further reduced (Supplementary Fig. 3d-f) We have added these data into Supplementary Fig. 3 and manuscript as follows:

Lines 149-155 in revised manuscript – To confirm functional H₁R expression in MS cholinergic neurons, we compared the excitability of MS cholinergic neurons bathed in either ACSF or H₁R antagonist mepyramine, and found that MS cholinergic neurons showed lower response to the H₁R antagonist mepyramine (Supplementary Fig. 3a-c). In addition, low *Hrh1* mRNA following knockdown showed low intrinsic excitability, and the response of MS cholinergic neurons to the H₁R antagonist mepyramine was further reduced (Supplementary Fig. 3d-f)

Supplemental figure 3. The functional H₁R expression in MS cholinergic neurons at baseline and the effective knockdown by miRNA. (a-b and d-e) Threshold current to elicit action potential with the increase of injected currents in MS cholinergic neurons bathed with ACSF or H₁R antagonist mepyramine. **(c and f)** Spike numbers with the increase of injected currents in MS cholinergic neurons bathed with ACSF or H₁R antagonist mepyramine. ***P*<0.01, ****P*<0.001. Data are represented as mean ± SEM.

3- The authors studied the brain regions downstream of MS-Chat that support the effect of HR1. They examined cFos expression after retrieval, and found a selective increase in DG granule cells of cKO mice.

Without a group of mice trained but not re-exposed, it is not possible to know what regions were specifically recruited by retrieval. It is also desirable to know if increased DG activation was paralleled by reduced cFos of MS-Chat cells in cKO, as implied from their circuit model. Finally, a statistical comparison of main Fig.3 and supplementary Fig.3 would help identify the impact of cKO on behavioral activation.

Response: Thanks for this constructive comment. We examined the c-Fos expression of *ChAT-Cre* and *ChAT-Cre;Hrh1^{fl/fl}* mice trained but not re-exposed. We found that there were no significant differences in the c-fos expression between *ChAT-Cre* and *ChAT-Cre;Hrh1^{fl/fl}* mice, which suggested that DG was specifically recruited by retrieval (**Supplemental figure 4**). We also conducted a statistical comparison of main Fig.3 and supplementary Fig.3 (Figure 3c in revised manuscript). In addition, there is no statistical difference of the MS c-fos density between *ChAT-Cre* and *ChAT-Cre;Hrh1^{fl/fl}* mice after contextual fear retrieval. To better address reviewer's concern, we conducted additional experiments. The AAV-Ef1 α -DIO-axon-GCaMP6s was microinjected in MS of *ChAT-Cre* and *ChAT-Cre;Hrh1^{fl/fl}* mice, and optical cannula were implanted in DG for fiber photometry. We found that the fluorescent signal of MS^{Ach}-DG axons of *ChAT-Cre* mice immediately decreased before the onset of freezing bouts and increased before the onset of mobility bouts, while *ChAT-Cre;Hrh1^{fl/fl}* mice showed a flat trace, which is consistent with the increase or decrease of Ach release during the contextual fear retrieval phase. The results illustrated the functional connection between MS^{Ach}-DG modulated by H₁R during the contextual fear retrieval phase. We have added these data into Supplementary Fig. 3 and Figure 4n-y and manuscript as follows:

Lines 176-180 in revised manuscript – We also compared the c-fos expression of *ChAT-Cre* and *ChAT-Cre;Hrh1^{fl/fl}* mice trained but not re-exposed, and found that there were no significant differences in the c-fos expression of *ChAT-Cre* and *ChAT-Cre;Hrh1^{fl/fl}* mice (Supplementary Fig. 4a-b). The statistical results indicate that DG was specifically recruited by the retrieval of contextual fear memory (Fig. 3c).

Lines 232-241 in revised manuscript – We next proved that whether MS^{Ach}-DG are functionally connected during the contextual fear retrieval phase, AAV-Ef1 α -DIO-axon-GCaMP6s was expressed in MS of *ChAT-Cre* and *ChAT-Cre;Hrh1^{fl/fl}* mice, and optical cannula were implanted in DG for fiber photometry (Fig. 4n-p). The fluorescent signal of MS^{Ach}-DG axons of *ChAT-Cre* mice immediately decreased before the onset of freezing bouts and increased before the onset of mobility bouts, while *ChAT-Cre;Hrh1^{fl/fl}* mice showed flat trace,

which is consistent with the increase or decrease of Ach release during the contextual fear retrieval phase (Fig. 4q-y). The results illustrated the functional connection between MS^{ACh} -DG modulated by H_1R during the contextual fear retrieval phase.

Supplemental figure 4. The c-Fos expression of *ChAT-Cre* and *ChAT-Cre;Hrh1^{fl/fl}* mice trained but not re-exposed. (a) Representative images of c-fos expressing neurons in the amygdala, medial prefrontal cortex and hippocampus of *ChAT-Cre* and *ChAT-Cre;Hrh1^{fl/fl}* mice trained but not re-exposed. Left panel: scale bar = 20 μ m. Right panel: scale bar = 30 μ m. (b) The percentage of c-fos expressing neurons of the indicated areas. n=3 mice per group. ns, not significant. Data are represented as mean \pm SEM.

Figure 3. The DG is the major downstream region of MS cholinergic neurons responsible for enhanced contextual fear memory in *ChAT-Cre;Hrh1^{fl/fl}* mice. (a) Representative images of c-fos expressing neurons in the amygdala, medial prefrontal cortex and hippocampus of *ChAT-Cre* and *ChAT-Cre;Hrh1^{fl/fl}* mice at 1.5 h after contextual fear memory retrieval. Scale bar = 30 μ m. n = 5 mice per group. (b) The percentage of c-fos expressing neurons of the indicated areas. (c) The percentage of c-fos expressing neurons in DG region of *ChAT-Cre* and *ChAT-Cre;Hrh1^{fl/fl}* mice re-exposed or not. (d) The curve of freezing level during the exploration period and each trial on the conditioning day for (a) and (b). (e) The curve of freezing level in each minute (left panel) and the percentage of freezing time during the contextual fear memory retrieval (right panel) for (a) and (b).

Figure 4n-y. (n) Representative image of AAV-DIO-axon-GCaMP6s expression in MS and DG. Left and middle panel: scale bar = 100 μ m. Right panel: scale bar = 20 μ m. (o) The curve of freezing level during the exploration period and each trial on the conditioning day for (r-y). (p) The curve of freezing level in each minute (left panel) and the percentage of freezing time during the contextual fear memory retrieval (right panel) for (r-y). (q) Example trace of the fluorescence signal of MS projecting axons in DG and behavioral epochs during the contextual fear retrieval. Blue boxes above the traces indicate freezing bouts (light blue < 3 s; dark blue \geq 3 s). Yellow boxes above the traces indicate mobility bouts (light yellow < 3 s; dark yellow \geq 3 s). (r and v) Averaged plots of fluorescence signal of MS projecting axons in DG aligned to the onset of freezing bouts of both *ChAT-Cre* mice and *ChAT-Cre;Hrh1^{fl/fl}* mice. (s-t and w-x) Heatmap representations of fluorescence signal of MS projecting axons in DG aligned to the onset of freezing bouts of *ChAT-Cre* mice (s and w) and *ChAT-Cre;Hrh1^{fl/fl}* mice (t and x). Color bars at the right of each heatmap represent different individual mice. (u and y) Normalized AUC of fluorescence signal of MS projecting axons in DG for freezing bouts. ** $P < 0.01$, *** $P < 0.001$, ns, not significant. Data are represented as mean \pm SEM.

4-The authors used retrograde tracing with an inducible, activity-dependent viral vector to label MS cells that project to active DG cells during retrieval. They found that about half of MS afferents were Chat(+). Please clarify the statistical test performed here and purpose (panel 3F).

Response: Thanks for this comment. Consistent with previous study^{12,13}, the retrograde tracing with an inducible, activity-dependent viral vector indicated that retrieval-induced neurons mediate a structural connection from the MS to the DG. Combined with the major concern e) from reviewer #2, we found half of contextual fear retrieval-induced neurons in DG was retrogradely connected with MS cholinergic neurons (53.3% of rabies virus [RV]-transfected cells were ChAT⁺ cells), whereas cued fear retrieval-induced neurons in DG' retrograde tracing resulted in minimal MS cholinergic labeling (11.1% of RV-transfected cells were ChAT⁺ cells), suggesting the MS-DG cholinergic neural circuitry engaged with high probability when contextual fear retrieval is occurring. We have added these data into Fig.3 as follows:

Figure 3. The DG is the major downstream region of MS cholinergic neurons responsible for enhanced contextual fear memory in *ChAT-Cre;Hrh1^{fl/fl}* mice. (f) The schematic for the strategy to retrogradely label MS cholinergic neurons that project to retrieval-induced neurons in DG. (g and k) The timeline of retrograde transsynaptic tracing experiment. (h and l) Left panel: Representative images of EGFP and mCherry double-labeled starter cells in the DG. Upper panel: scale bar = 50 μ m. Lower panel: scale bar = 20 μ m. Right

panel: Representative images show ChAT (the marker of cholinergic neurons) and EGFP (expressed by retrogradely labeled cells) expression in the MS. Left panel: scale bar = 100 μ m. Right panel: scale bar = 20 μ m. Arrows indicate EGFP and ChAT doublelabeled cells. **(i and m)** Upper panel: Quantification of rabies-labeled MS neurons. Lower panel: Percentage of rabies-labeled, immunochemically identified MS cholinergic neurons. n=3 mice. **(j)** Left panel: Representative images of c-fos expressing neurons in the DG at 1.5 h after cued fear memory retrieval. Scale bar = 50 μ m. Right panel: The percentage of c-fos expressing neurons of the indicated areas. n = 3 mice per group. * P <0.05, ** P <0.01, ns, not significant. Data are represented as mean \pm SEM.

5-Fiber photometry of the acetylcholine reporter Ach3 was used to assess the impact of HR1 deletion in Chat cells on Ach levels in the DG. An inverse relationship between Ach levels and freezing was observed. Ach fluctuations were largely ablated in cKO mice, indicating that HR1 in Chat cells facilitates Ach release during re-exposure to block memory expression.

However, the authors showed that Hrh1 deletion in Chat cells reduces the spike probability in response to depolarization. Because conditioning itself persistently decreased Hrh1 expression in WT, it is not clear then how Ach would be preferentially released during retrieval. It is therefore important to (a) define the cell type undergoing Hrh1 downregulation after training, (b) test if after conditioning the rheobase of MS-Chat cells increases, and (c) discuss this potential conundrum.

Response: Thanks for this constructive comment. We totally agree with reviewer's concern about the contradictory relationship between the enhancement of MS-DG cholinergic signals in fear memory retrieval and the physiological decrease of MS H₁R after fear training, which may lead to a decrease in excitability of MS cholinergic neurons. Therefore, to explain more clearly, we added several experiments point-to-point firstly, and found that (a): at different time after fear training (1,7,14,28 days), H₁R selectively decreased expression on MS cholinergic rather than GABAergic or glutamatergic neurons. (b): And its rheobase increased significantly from day 7-day 28, which consistent with the trend of *Chat-Cre;Hrh1^{fl/fl}* mice. However, it can be observed that on day 1 after fear training, rheobase of trained group did not show significant changes compared to the control group, and even when H₁R physiology decreased to 28 days, the increase in rheobase of trained group (171.4%, Figure 2c and d) did not reach the level of *Chat-Cre;Hrh1^{fl/fl}* group (403.8%, Figure 2m and n). Then we added discussion in the revised manuscript:

Lines 402-423 in revised manuscript – These results indicate that H₁R in MS cholinergic neurons facilitates Ach release during re-exposure to suppress memory expression. However, we also found that both artificially knocking out or physiologically decreasing H₁R expression on MS cholinergic neurons lead

to an increase rheobase of MS cholinergic neurons (Figure 2). Therefore, it seems puzzling that the H₁R expression on MS cholinergic neurons, which selectively continuously decreases after fear conditioning, is specifically responsible for activating the MS^{ACh}→DG circuit in fear memory retrieval. We provide some possible explanations based on experimental results and previous works. Firstly, we observed that in the paradigm of fear memory retrieval in our study, although H₁R expression on MS cholinergic neurons significantly decreased after 1 day of conditioned fear training, their rheobase did not showed significantly change, suggesting that MS cholinergic function may not have undergone a physiological decline. But why does the MS cholinergic signal show a specific increase in fear memory retrieval? Considering the significant ablation of this function after depletion of H₁R in MS cholinergic neurons, we speculate that the specifically enhanced cholinergic signals in fear memory retrieval may be mediated by upstream histaminergic neural circuits. In our previous study on feeding circuits, we found that the TMN-MS histaminergic neural circuit specifically acts on MS glutamatergic, rather than GABAergic or cholinergic neurons¹⁴. Therefore, we speculate that there may also be a histaminergic neural circuit specifically acting on MS cholinergic rather than GABAergic or glutamatergic neurons in the fear memory retrieval. And this study is also included in our future research plans.

Figure 2. H₁R in MS cholinergic neurons is critical for contextual fear memory. (a) Representative images of RNAscope in situ hybridization of *Hrh1* mRNA together with choline acetyltransferase (ChAT), *VGAT* mRNA and *VGLUT2* mRNA in the MS after contextual fear conditioning. Scale bar = 20 μ m.

(b) Quantitative analysis of *Hrh1* mRNA expression in ChAT⁺, VGAT⁺ and VGLUT2⁺ cell. (c and d) Threshold current to elicit action potential with the increase of injected currents in MS cholinergic neurons recorded by whole-cell patch-clamp at 1 day, 7 days, 14 days and 28 days post-contextual fear memory retrieval.

6- At the circuit level, their data imply that Ach in DG blocks granule cell activation, and that (excess) DG activation in cKO impairs memory recall. They confirmed that chemogenetically activating MS-Chat terminals in the DG of cKO mice, or inhibiting granule cells themselves, normalizes (i.e. decreases) memory. In contrast, optogenetic activation of all MS cells projecting to DG during the retrieval test increases memory. Such opposite effects of MS-Chat→DG activation and MS-pan→DG activation should be directly discussed (e.g. different HR1 expression, diverse MS populations, etc). Also, please state in the main text remote vs recent memory tests in these last experiments. In a final experiment, the authors found that an agonist of Gi-coupled M4Rs infused in DG reduced memory in cKO mice, providing a plausible mechanism for the effect of HR1 in DG inhibition and memory.

Response: Thank you very much for these comments. (1) As reviewer commented, we also found the opposite effects of MS-Chat→DG activation and MS-pan→DG activation on fear memory recall. Therefore, we have added a discussion on this issue:

Lines 372-388 in revised manuscript – However, it can be found that MS^{Ach}→DG activation showed an opposite effect on fear memory recall (Figure 5) compared to MS→DG activation (Supplementary Figure 7), which seems contradictory. Firstly, the MS nucleus is recognized to be involved in the regulation of various memory functions, including fear memory^{2,15,16}. But it has three different types of neurons, as known as cholinergic, GABAergic, and glutamatergic neurons, and there are many works proving its differential regulatory role in memory functions¹⁷⁻¹⁹. Therefore, we speculate that the difference in circuit function between MS^{Ach}→DG and MS→DG may be determined by the specific function of diverse MS subpopulations. In addition, in our previous study on the functional feeding circuit, we found that the upstream histaminergic neural projection specifically acts on MS glutamatergic (but not GABAergic or cholinergic) neurons and exerts an inhibitory effect on mouse feeding behavior¹. And we speculate that the H₁R on different neurons of MS may also receive differential effects from upstream histaminergic neural circuits and be responsible for different regulatory functions in fear memory retrieval. Therefore, there are multiple possible mechanisms of the opposite effects of MS^{Ach}→DG activation and MS→DG activation, and further exploration is needed in the future study.

(2) We used a recent memory test for the last experiment in the main text. We apologize for not writing it clearly in the original manuscript. And we have added corresponding descriptions in the revised manuscript:

Lines 292-304 in revised manuscript – We also explored whether the inhibitory muscarinic M4 receptors (M4Rs) were involved in the enhanced recent contextual fear retrieval circuit induced by H1R deficiency in MS cholinergic neurons. The muscarinic M4R antagonist tropicamide, M4R agonist MCN-A-343 or vehicle was bilaterally micro-infused into the DG through cannulas 30 min before contextual fear memory retrieval (Supplementary Fig. 10a-b). The vehicle and drug-treated mice showed comparable levels of fear learning and cued fear memory (Supplementary Fig. 10c and e). Importantly, MCN-A-343 administration in the DG effectively reversed enhanced recent contextual fear memory, while tropicamide group exhibited slightly stronger recent contextual fear memory compared to the vehicle group (Supplementary Fig. 10d). These results indicate that M4R in the DG is responsible for the regulation of recent contextual fear memory and is the downstream effector of the enhanced contextual fear retrieval circuit induced by H1R deficiency in MS cholinergic neurons.

Reviewer #2 (Remarks to the Author):

Chen and colleagues provide a well-presented manuscript containing a novel line of inquiry that investigates the contribution of the H1 histamine receptor (H1R) within the medial septum (MS) to dentate gyrus (DG) pathway during expression of context fear behavior. The authors use the multiple techniques of transgenic mice, fiber photometry, chemogenetics, ex vivo electrophysiology, optogenetics, neuropharmacology, and intersectional AAV expression to build the case that the absence of H1R in MS cholinergic neurons projecting to the DG enhances, specifically, context fear expression/retrieval. They show that *ChAT-Cre;Hrh1* mice show reduced excitability (rheobase) of ACh MS neurons. Furthermore, cholinergic MS projecting to context fear memory engram cells in the DG, and the cholinergic activity of *ChAT-Cre;Hrh1*, is reduced during context fear expression, as measured by fiber photometry. Finally, using chemogenetic methods the authors demonstrate that activating the MS-DG pathway with CNO reduces the elevated context fear phenotype in *ChAT-Cre;Hrh1* mice. The claims of the authors are highly specific regarding neuronal circuitry and behavior; moreover, investigating the role of histamine receptors in the expression of defensive behavior is a novel and important advance to the field. However, there are several limitations in the current form of the manuscript that reduce this reviewer's enthusiasm. Below are suggestions to be addressed prior to publication.

Response: Thank you very much for reviewing our work and propounding valuable comments, which are helpful for us to improve the quality of our paper significantly. Below are our point-by-point responses to the comments.

Major concerns:

a) The behavioral results in Fig 1 are strong and convincing. However, since the effects of H1R on cued and context fear is a novel line of inquiry, the authors should demonstrate that the specificity of elevated context fear expression is not a consequence of testing order effects, i.e., a control study that tests cued fear first then four hours later tests context fear is required.

Response: Thanks for this important comment. We totally agree with the reviewer that the potential impact of testing order effects on the specificity of elevated context fear expression should be confirmed. According to the suggestion, we have added control study that tests cued fear first then four hours later tests context fear. We still found that *ChAT-Cre;Hrh1^{fl/fl}* mice exhibit enhanced contextual fear memory, yet normal fear learning and cued fear memory, which was consistent with previous data (Fig 1 a-g in the original manuscript). These findings showed that the specificity of elevated context fear expression in *ChAT-Cre;Hrh1^{fl/fl}* mice is not a consequence of testing order effects. We have added these data into Fig.1h-n and manuscript as follows:

Lines 97-104 in revised manuscript – To exclude the effect of testing order on the specificity of elevated context fear expression, we conducted experiments that cued fear memory was tested first, followed by contextual fear memory test four hours later. Consistent with the behavior in the opposite testing order, *ChAT-Cre;Hrh1^{fl/fl}* mice still exhibited enhanced contextual fear memory while displaying normal fear learning and cued fear memory (Fig 1 h-n). These findings showed that the specificity of elevated context fear expression in *ChAT-Cre;Hrh1^{fl/fl}* mice is not a consequence of testing order effects.

Figure 1h-n. Decreased histamine H₁R expression in cholinergic neurons selectively enhances contextual fear memory. (h) Schematic diagram of the experimental design of recent and remote fear conditioning test. (i and l) The curve of freezing level during the exploration period and each trial on the conditioning day. (j and m) The curve of freezing level in each minute (left panel) and the percentage of freezing time during the contextual fear memory retrieval (right panel). (k and n) The percentage of freezing time during the cued fear memory retrieval. n=8-10 mice per group. * $P < 0.05$, ** $P < 0.01$, *** $P < 0.001$, ns, not significant. Data are represented as mean \pm SEM.

b) The importance of the MS-DG pathway is a central feature of this manuscript, for this reason understanding the relationship between DG activation and context freezing behavior is essential. The addition of context freezing behavior for control and *ChAT-Cre;Hrh1* mice used in Fig 3a-b would further support the specificity of DG activation level (as measured by c-fos in Fig 3) and heightened context freezing behavior.

Response: Thank you very much for this valuable comment. We have added these data into Fig.3d and 3e as follows:

Figure 3. The DG is the major downstream region of MS cholinergic neurons responsible for enhanced contextual fear memory in *ChAT-Cre;Hrh1^{fl/fl}* mice. (a) Representative images of c-fos expressing neurons in the amygdala, medial prefrontal cortex and hippocampus of *ChAT-Cre* and *ChAT-Cre;Hrh1^{fl/fl}* mice at 1.5 h after contextual fear memory retrieval. Scale bar = 30 μ m. n = 5 mice per group. (b) The percentage of c-fos expressing neurons of the indicated areas. (c) The percentage of c-fos expressing neurons in DG region of *ChAT-Cre* and *ChAT-Cre;Hrh1^{fl/fl}* mice re-exposed or not. (d) The curve of freezing level during the exploration period and each trial on the conditioning day for (a) and (b). (e) The curve of freezing level in each minute (left panel) and the percentage of freezing time during the contextual fear memory retrieval (right panel) for (a) and (b).

c) The use of fiber photometry and the Ach3.0 is a strong contribution to this manuscript. However, some questions remain for the recordings and analyses in Fig 4. Panels g and k collapse freezing trials across all mice and statistically compare “area under the curve”; however, the robustness of the effects in panels d-k between mouse subjects would be supported by an analysis that compared area under the curve for individual *ChAT-Cre;Hrh1* and *ChAT-Cre* mice (i.e., take the average dF/F per mouse and compare between *ChAT-Cre;Hrh1* and *ChAT-Cre* groups). Providing the context freezing levels of the mice in this fiber photometry experiment would also help support the relationship between MS ACh in the DG and normal or heightened context fear behavior. Finally, it is unclear if only freezing bouts greater than 3 seconds were considered in this analysis.

Response: Thanks for the important comments. As reviewer suggested, we reanalyzed the average dF/F of individual mouse between *ChAT-Cre* and *ChAT-Cre;Hrh1^{fl/fl}* groups and found that the overall acetylcholine release in DG of *ChAT-Cre* mice was stronger than that in the *ChAT-Cre;Hrh1^{fl/fl}* mice during the retrieval phase of contextual fear memory (Fig 4i and 4m in the revised manuscript). The same statistical comparison was used for AAV-DIO-axon-GCaMP6 experiments (Fig 4u and 4y in the revised manuscript). We also provided the context freezing levels of the mice in this fiber photometry experiment (Fig 4c&d and Fig 4o&p in the revised manuscript). In addition, all the freezing bouts (rather than only bouts greater than 3 seconds) were considered in this analysis. We have added these data into Fig.4 as follows:

Figure 4. The *ChAT-Cre;Hrh1^{fl/fl}* mice exhibit functional acetylcholine deficiency of MS-DG circuit during the retrieval. (a) Schematic illustrating the fiber photometry setup for recording acetylcholine release during the contextual fear retrieval. (b) Illustration of the viral injection site, viral expression, and optic fiber placement (indicated by the white dotted line) in the DG. Scale

bar = 500 μ m. **(c)** The curve of freezing level during the exploration period and each trial on the conditioning day for (f-m). **(d)** The curve of freezing level in each minute (left panel) and the percentage of freezing time during the contextual fear memory retrieval (right panel) for (f-m). **(e)** Example trace of acetylcholine release in the DG and behavioral epochs during the contextual fear retrieval. Blue boxes above the traces indicate freezing bouts (light blue < 3 s; dark blue \geq 3 s). Yellow boxes above the traces indicate mobility bouts (light yellow < 3 s; dark yellow \geq 3 s). **(f and j)** Averaged plots of acetylcholine release aligned to the onset of freezing bouts of both *ChAT-Cre* mice and *ChAT-Cre;Hrh1^{fl/fl}* mice. **(g-h and k-l)** Heatmap representations of acetylcholine release aligned to the onset of freezing bouts of *ChAT-Cre* mice **(g and k)** and *ChAT-Cre;Hrh1^{fl/fl}* mice **(h and l)**. Color bars at the right of each heatmap represent different individual mice. **(i and m)** Normalized AUC of DG acetylcholine release for freezing bouts. **(n)** Representative image of AAV-DIO-axon-GCaMP6s expression in MS and DG. Left and middle panel: scale bar = 100 μ m. Right panel: scale bar = 20 μ m. **(o)** The curve of freezing level during the exploration period and each trial on the conditioning day for (r-y). **(p)** The curve of freezing level in each minute (left panel) and the percentage of freezing time during the contextual fear memory retrieval (right panel) for (r-y). **(q)** Example trace of the fluorescence signal of MS projecting axons in DG and behavioral epochs during the contextual fear retrieval. Blue boxes above the traces indicate freezing bouts (light blue < 3 s; dark blue \geq 3 s). Yellow boxes above the traces indicate mobility bouts (light yellow < 3 s; dark yellow \geq 3 s). **(r and v)** Averaged plots of fluorescence signal of MS projecting axons in DG aligned to the onset of freezing bouts of both *ChAT-Cre* mice and *ChAT-Cre;Hrh1^{fl/fl}* mice. **(s-t and w-x)** Heatmap representations of fluorescence signal of MS projecting axons in DG aligned to the onset of freezing bouts of *ChAT-Cre* mice **(s and w)** and *ChAT-Cre;Hrh1^{fl/fl}* mice **(t and x)**. Color bars at the right of each heatmap represent different individual mice. **(u and y)** Normalized AUC of fluorescence signal of MS projecting axons in DG for freezing bouts. ** $P < 0.01$, *** $P < 0.001$, ns, not significant. Data are represented as mean \pm SEM.

d) Interpretation of Fig 5 requires the addition of control groups to assess the elevated context fear in *ChAT-Cre;Hrh1* (Fig 5d) and miR30shRNA knockdown (Fig. 5i) by adding the *ChAT-Cre* group (or a genotype that will express “normal” levels of context fear). This will support the claim that the DREADD manipulations restored context freezing behavior to “normal” levels.

Response: We are sorry for the careless experimental design. As reviewer suggested, groups that express “normal” levels of context fear were added in Fig.5 as follows:

Figure 5. H₁R in DG-projecting cholinergic neurons in the MS is essential for the retrieval of contextual fear memory. (a) Left panel: representative images of the hM3Dq expression in the MS. The percentage of mCherry+ cells co-expressing ChAT and percentage of ChAT+ cholinergic neurons co-expressing mCherry in the MS were quantified. Scale bar = 20 μ m. *n* = 3 mice per group. Right panel: representative image of cannula placement (indicated by the white dotted line) in the DG. Scale bar = 500 μ m. (b) The timeline of chemogenetic activation experiment. (c and h) The curve of freezing level during the exploration period and each trial on the conditioning day. (d and i) The curve of freezing level in each minute (left panel) and the percentage of freezing time during the contextual fear memory retrieval (right panel). (e and j) The percentage of freezing time during the cued fear memory retrieval. *n* = 10 mice per group. (f) Left panel: representative images of the AAV-miR30shRNA(Hrh1) expression in the MS. Scale bar = 30 μ m. Right panel: representative images show the absence of colocalization between c-fos and mCherry in DG after CNO administration. Scale bar = 30 μ m. (g) Experimental scheme for manipulation of H₁R in MS-DG circuit. *n* = 12 mice per group. **P* < 0.05, ***P* < 0.01, ****P* < 0.001, *ns*, not significant. Data are represented as mean \pm SEM.

e) The specificity of the MS-DG context vs. cued fear retrieval circuitry would be further supported by the inclusion of a cued fear memory retrieval group that undergoes the retrograde and engram tagging procedure in Fig 3c-f. Because all mice undergo fear conditioning with 3 tone cue & shock pairing, they are all encoding the tone-shock association. However, the specificity of the context vs cued freezing effect depends on the MS-DG circuitry being engaged only when context retrieval is occurring. For this reason, a quantification of cell activated during cued fear retrieval within the MS-DG pathways (as shown in Fig 3f) is necessary.

Response: Thank you very much for this valuable comment. Accordingly, we performed the experiment (as shown in Fig 3c-f in the original manuscript) to quantify the activated cell during cued fear retrieval within the MS-DG pathways. We found that a small part of MS cholinergic neurons (11.1% of RV cells were ChAT positive cells) sent ascending monosynaptic inputs to cued fear retrieval-induced neurons in DG, which was much less than the contextual fear retrieval group (53.3% of RV cells were ChAT positive cells). Our results suggested the MS-DG cholinergic neural circuitry is highly engaged during context retrieval is occurring. We have added these data into Fig.3 as follows:

Figure 3. The DG is the major downstream region of MS cholinergic neurons responsible for enhanced contextual fear memory in *ChAT-Cre;Hrh1^{fl/fl}* mice. (f) The schematic for the strategy to retrogradely label MS cholinergic neurons that project to retrieval-induced neurons in DG. (g and k) The timeline of retrograde transsynaptic tracing experiment. (h and l) Left panel: Representative images of EGFP and mCherry double-labeled starter cells in the DG. Upper panel: scale bar = 50 μ m. Lower panel: scale bar = 20 μ m. Right panel: Representative images show ChAT (the marker of cholinergic neurons) and EGFP (expressed by retrogradely labeled cells) expression in the MS. Left panel: scale bar = 100 μ m. Right panel: scale bar = 20 μ m. Arrows indicate EGFP and ChAT double-labeled cells. (i and m) Upper panel: Quantification of rabies-labeled MS neurons. Lower panel: Percentage of rabies-labeled, immunohistochemically identified MS cholinergic neurons. n=3 mice. (j) Left panel: Representative images of c-fos expressing neurons in the DG at 1.5 h after cued fear memory retrieval. Scale bar = 50 μ m. Right panel: The percentage of c-fos expressing neurons of the indicated areas. n = 3 mice per group. *P<0.05, **P<0.01, ns, not significant. Data are represented as mean \pm SEM.

Minor concerns:

a) Consistent use of “consolidation” period and “conditioning” session would help clarify when fiber photometry (and other manipulations) are being implemented.

Response: Thanks for this constructive comment. Accordingly, we have proofed our manuscript carefully. The example sentences have been revised as follows:

Lines 89-91 in revised manuscript – Compared with control mice, *ChAT-Cre;Hrh1^{fl/fl}* male mice displayed similar freezing levels across trials during the conditioning session, suggesting the normal ability in fear learning (Fig. 1b and e).

Lines 207-210 in revised manuscript –Because we found that decreased H₁R in cholinergic neurons had no effect on conditioning, the acetylcholine release was recorded 30 min after conditioning (the consolidation phase) or during contextual fear memory test (the retrieval phase).

Lines 279-281 in revised manuscript - However, intracranial microinjection of CNO immediately after conditioning has no effect on the behaviors of the *ChAT-Cre;Hrh1^{fl/fl}* mice (Supplementary Fig. 9).

b) Additional details are required in the methods section that specify the product numbers for antibodies used (e.g., cFos, secondary, DAPI, ChAT).

Response: Thank you very much for this comment. As suggested, we have added additional details about the product numbers for antibodies or RNAscope™ probes in the methods section.

c) The abstract mentions “reciprocally connected” MS ACh neurons and DG cells, but I did not see an investigation of DG projections to MS ACh neurons. Alternative word choice is recommended.

Response: Thank you very much for this constructive comment. Indeed, we are truly aware of our mistake that mentions “reciprocally connected” MS ACh neurons and DG cells in the abstract. As suggested, we have corrected the statement in the revised manuscript as following:

Lines 25-28 in revised manuscript - Importantly, contextual fear memory induces differential activation of the dentate gyrus (DG) neurons in *ChAT-Cre;Hrh1^{fl/fl}* mice, and the activated DG neurons received projections from the MS cholinergic neurons.

d) Freezing was quantified with PANLAB software, but the criterion for freezing behavior is not stated. Please add the freezing criteria.

Response: We thank reviewer for the constructive suggestion. The freezing criteria has been added in the revised manuscript. Please see method part, lines 518-519.

e) Line 341 mentions a “behavioral closed-loop” chemogenetic manipulation. The chemogenetics manipulations did not appear to be “closed-loop”; please consider re-wording or expanding on why the use of DREADDs was closed-loop.

Response: We are very sorry for the inappropriate description. According to reviewer’s suggestion, we have removed the incorrect description in the revised manuscript.

f) Please add labels to the color bars in figure 4.

Response: We are sorry for this mistake. Labels to the color bars in figure 4 have been added in the revised manuscript.

g) The discussion could include more information about H1R physiology in the MS and how this receptor is regulated by experiential factors like fear conditioning.

Response: Thank you very much for this valuable comment. Combined with the comments from reviewer #1, we have added comments in the revised discussion as follows:

Lines 457-468 in revised manuscript – Histaminergic system in the brain plays a vital role in many pathophysiological processes, especially learning and memory, and has been proposed to be involved in the neuropathology of fear- and emotion-related disorders. In addition, histamine release has been proposed to be a sensitive indicator of stress. For example, histamine turnover rates in the brain are increased in response to stressful stimulation, including the application of electrical shocks or chronic restraint stress. Our recent study have shown that H₃R in the vBF cholinergic neurons is responsible for the regulation of contextual fear memory by histamine, suggesting the significant interaction between histaminergic and cholinergic systems in contextual fear memory. As the important member of histamine receptors, H₁R has been found to modulate the activity of cholinergic septal neurons by regulation of its depolarization. However, the precise role of H₁R in cholinergic neurons is still not been clarified.

h) The introduction and discussion should include additional information about the muscarinic receptor to integrate the results in Supp Fig 7 with the rest of the manuscript’s results.

Response: Thank you very much for this comment. As suggested, we have additional information about the muscarinic receptor as follows:

Lines 426-450 in revised manuscript – Although there is some inconsistency in the literature, pharmacological interventions with muscarinic agonists or antagonists support the general premise that muscarinic acetylcholine receptor (mAChR) in the hippocampus appears particularly critical in the acquisition of contextual fear learning during the acquisition and/or consolidation of fear learning. Several studies have indicated that the systemic administration of the scopolamine (mAChR antagonist) before or after training can reduce the acquisition and retrieval of conditioned fear. However, some other studies failed to see effects of scopolamine or dicyclomine (M1 mAChR antagonist) given after the training session on contextual fear responses even with high doses, suggesting the role of mAChR in fear memory progress is critical and intricate¹. Accumulating evidence suggests that selective M4 mAChR may offer a novel target for the treatment of psychosis related to cognitive impairments. VU0152100 (a highly selective M4 positive allosteric modulator) blocks amphetamine-induced disruption of the acquisition of contextual fear conditioning⁴ and retrieval of remotely acquired contextual memories requires retrosplenial cortex M4 mAChR activity⁵. Further, study using a combination of optogenetic techniques, in vivo and in vitro electrophysiology and multiphoton imaging showed that acetylcholine release from cholinergic septohippocampal projections can cause slow inhibition of hippocampal dentate granule cells via astrocyte intermediaries⁶. By regulating granule cell excitability and the size of DG memory ensembles, the cholinergic input from the MS to the hilar perforant path-associated (HIPP) cells in DG control background context fear⁷. Our study indicate that the inhibitory muscarinic M4R in the DG is the downstream effector of the enhanced contextual fear retrieval circuit induced by H1R deficiency in MS cholinergic neurons, which further highlights the important role of M4R in contextual fear memory.

i) Based on the limited amount of data that explores the 28-day remote time point of context memory fear retrieval, the amount of discussion on systems consolidation results should be tempered.

Response: Thanks for the great suggestion. We have appropriately tempered the amount of discussion on systems consolidation results. Please see discussion part.

Reference

1. Knox, D. The role of basal forebrain cholinergic neurons in fear and extinction memory. *Neurobiol Learn Mem* **133**, 39-52 (2016).
2. Li, X., *et al.* Molecularly defined and functionally distinct cholinergic subnetworks. *Neuron* **110**, 3774-3788.e3777 (2022).
3. Bloem, B., *et al.* Topographic mapping between basal forebrain cholinergic neurons and the medial prefrontal cortex in mice. *J Neurosci* **34**, 16234-16246 (2014).
4. Byun, N.E., *et al.* Antipsychotic drug-like effects of the selective M4 muscarinic acetylcholine receptor positive allosteric modulator VU0152100.

- Neuropsychopharmacology* **39**, 1578-1593 (2014).
5. Leaderbrand, K., *et al.* Muscarinic acetylcholine receptors act in synergy to facilitate learning and memory. *Learn Mem* **23**, 631-638 (2016).
 6. Pabst, M., *et al.* Astrocyte Intermediaries of Septal Cholinergic Modulation in the Hippocampus. *Neuron* **90**, 853-865 (2016).
 7. Raza, S.A., *et al.* HIPP neurons in the dentate gyrus mediate the cholinergic modulation of background context memory salience. *Nat Commun* **8**, 189 (2017).
 8. Ito, C. The role of brain histamine in acute and chronic stresses. *Biomed Pharmacother* **54**, 263-267 (2000).
 9. Westerink, B.H., *et al.* Evidence for activation of histamine H3 autoreceptors during handling stress in the prefrontal cortex of the rat. *Synapse* **43**, 238-243 (2002).
 10. Zheng, Y., *et al.* Postsynaptic histamine H(3) receptors in ventral basal forebrain cholinergic neurons modulate contextual fear memory. *Cell Rep* **42**, 113073 (2023).
 11. Gorelova, N. & Reiner, P.B. Histamine depolarizes cholinergic septal neurons. *J Neurophysiol* **75**, 707-714 (1996).
 12. Wang, C., *et al.* Tactile modulation of memory and anxiety requires dentate granule cells along the dorsoventral axis. *Nat Commun* **11**, 6045 (2020).
 13. Wang, Y., *et al.* Direct Septum-Hippocampus Cholinergic Circuit Attenuates Seizure Through Driving Somatostatin Inhibition. *Biol Psychiatry* **87**, 843-856 (2020).
 14. Xu, L., *et al.* An H2R-dependent medial septum histaminergic circuit mediates feeding behavior. *Curr Biol* **32**, 1937-1948.e1935 (2022).
 15. Griguoli, M. & Pimpinella, D. Medial septum: relevance for social memory. *Front Neural Circuits* **16**, 965172 (2022).
 16. Boyce, R., Glasgow, S.D., Williams, S. & Adamantidis, A. Causal evidence for the role of REM sleep theta rhythm in contextual memory consolidation. *Science* **352**, 812-816 (2016).
 17. Wu, D., *et al.* Medial septum tau accumulation induces spatial memory deficit via disrupting medial septum-hippocampus cholinergic pathway. *Clin Transl Med* **11**, e428 (2021).
 18. Sans-Dublanc, A., *et al.* Septal GABAergic inputs to CA1 govern contextual memory retrieval. *Sci Adv* **6**(2020).
 19. Wu, X., Morishita, W., Beier, K.T., Heifets, B.D. & Malenka, R.C. 5-HT modulation of a medial septal circuit tunes social memory stability. *Nature* **599**, 96-101 (2021).

REVIEWER COMMENTS

Reviewer #1 (Remarks to the Author):

The authors have addressed all points of concern. The data in KO mice is logical and well articulated. Thus, deleting HR1 in ChAT+ MS cells limits their excitability, increases retrieval-induced DG activation, and increases memory retrieval. This effect correlates with the loss of Ach modulation in DG during re-exposure, causing deficits in the activation of M4R (an inhibitory AchR) in DG granule cells. The circuit-based approaches used to rescue or mimic the KO phenotype are convincing and specific.

What was less clear were the dynamics of HR1 levels and excitability in ChAT+ MS cells in WT mice after training. Despite the revision, the decrease of HR1 mRNA after training compared with naïve is still not evident (Fig.2a, first row). HR1 levels in ChAT+ control cells are hardly visible (row1, column 2). Second, the revised pharmacology does not show that ChAT neurons after training are less sensitive to HR1 antagonist either (Suppl.Fig.3), confirming concerns on the mRNA data. Third, it is still not shown if MS ChAT+ cells are recruited during retrieval (Fig.3). The authors claim in the rebuttal that their activity is not different between WT and HR1KO mice, without shown data, but both Ach release and axonal GCamP signals differ between genotypes (Fig.4). So how is that possible? [It remains unclear if photometry was done at recent (1-2 d?) or remote time points].

In short, Figs. 2a-d complicate the story line because the impact that these training-induced changes might have on memory retrieval in WT were not addressed.

One possibility is that those modest changes don't really have any impact in WT, allowing for proper modulation of Ach release during contextual memory. The authors imply instead that these changes will have an impact on memory, but that an unknown, hypothetical histaminergic input overrides them. I suggest they also consider the more parsimonious possibility above.

Reviewer #2 (Remarks to the Author):

The latest version of the manuscript is greatly improved and the authors have adequately addressed many of my concerns. There are only a few concerns / clarification remaining.

1) The addition of panels Fig3d,e helped to link the cFos histology with behavior, but the enhancement of context freezing in the ChAT-Cre;Hrh1 mice did not reach statistical significance ($p = 0.053$). Either performing additional cFos and behavior experiments are required to achieve statistical significance or adding text to the Results section that refers to the lack of statistical significance in panels Fig3d,e is required.

2) In the Methods or Results section, please specify the length of freezing bout considered for the DG and MS-DG fiber photometry quantification panels in Fig4. Providing a justification for using greater than 3 seconds in one set of analyses, but less than or greater than 3 sec in another set of analyses is required. Additionally, Fig4r-y appear to use a different data smoothing function compared to Fig4f-m. I was unable to find the details of smoothing parameters in the methods, please add these data processing details and justify if/why they were not applied similarly to the two fiber photometry data sets.

3) Regarding freezing criteria for PANLAB software, please specify the minimum duration for immobilization to be considered freezing.

Point-by-point response to reviewers' comments

REVIEWER COMMENTS

Reviewer #1 (Remarks to the Author):

The authors have addressed all points of concern. The data in KO mice is logical and well articulated. Thus, deleting HR1 in ChAT⁺ MS cells limits their excitability, increases retrieval-induced DG activation, and increases memory retrieval. This effect correlates with the loss of Ach modulation in DG during re-exposure, causing deficits in the activation of M4R (an inhibitory AchR) in DG granule cells. The circuit-based approaches used to rescue or mimic the KO phenotype are convincing and specific.

Response: We truly appreciate your consideration on our work and important comments. Our point-by-point responses are addressed below.

What was less clear were the dynamics of HR1 levels and excitability in ChAT⁺ MS cells in WT mice after training. Despite the revision, the decrease of HR1 mRNA after training compared with naïve is still not evident (Fig.2a, first row). HR1 levels in ChAT⁺ control cells are hardly visible (row1, column 2). Second, the revised pharmacology does not show that ChAT neurons after training are less sensitive to HR1 antagonist either (Suppl.Fig.3), confirming concerns on the mRNA data. Third, it is still not shown if MS ChAT⁺ cells are recruited during retrieval (Fig.3). The authors claim in the rebuttal that their activity is not different between WT and HR1KO mice, without shown data, but both Ach release and axonal GCaMP signals differ between genotypes (Fig.4). So how is that possible? It remains unclear if photometry was done at recent (1-2 d?) or remote time points.

Response: We thank reviewer for the constructive comments.

1) To make the expression of *Hrh1* mRNA in ChAT⁺ cells more clear, we adjusted all the images about *Hrh1* mRNA in MS ChAT⁺ cells with the same parameters. We observed that *Hrh1* mRNA expression in the MS cholinergic neurons significantly decreased at 1 day, 7 days, 14 days and 28 days post-contextual fear memory retrieval compared to controls. Please see revised Fig. 2a.

Revised Fig. 2a

Representative images of RNAscope *in situ* hybridization of *Hrh1* mRNA together with choline acetyltransferase (ChAT), VGAT mRNA and VGLUT2 mRNA in the MS after contextual fear conditioning. Scale bar = 20 μ m.

2) Using a similar approach as in panel 2m, we detected the response of MS ChAT⁺ cells to the H₁R antagonist at baseline or following selective knock-down of H₁R expression in ChAT⁺ cells. Our data showed that the excitability of MS cholinergic neurons bathed in H₁R antagonist was lower than that in ACSF at baseline (rheobase increased by 126.53%, Suppl.Fig.3a and b). The excitability of H₁R-knockdown cholinergic neurons bathed in H₁R antagonist was lower than that in ACSF (rheobase increased by 66.86%, Suppl.Fig.3d and e), suggesting cholinergic neurons with lower H₁R expression exhibited less sensitive to the H₁R antagonist when compared to baseline. Following your constructive comment, we have added related description in the revised manuscript. Please see lines 151-155.

Lines 151-155 in revised manuscript – “In addition, low *Hrh1* mRNA following knockdown showed low intrinsic excitability, and the response of MS cholinergic neurons to the H₁R antagonist Mepyramine was further reduced (Supplementary Fig. 3d-f), suggesting cholinergic neurons with lower H₁R expression exhibited less sensitive to the H₁R antagonist.”

3) In this study, no significant difference was observed about the MS c-Fos density between *ChAT-Cre* and *ChAT-Cre;Hrh1^{fl/fl}* mice after contextual fear

retrieval, while both Ach release and axonal GCaMP signals differ in DG between genotypes. This may be due to the fact that the extent of c-Fos activation may not show strictly positive correlation with the changes of the neuronal excitability and released neurotransmitter¹, and some minor or acute changes in neuronal excitability may be insufficient to lead to c-Fos expression changes in MS after contextual fear retrieval. Li et al. showed the patterned visual stimulation increased ACh released from HDB neurons, while the c-fos expression was unchanged within HDB neurons^{2,3}. In addition, the photometry was done at recent (1-2 d) time points. These experimental conditions have been added in the manuscript. Please see lines 210-213, lines 215-221 and lines 231-234.

Lines 210-213 in revised manuscript – “Briefly, an AAV expressing acetylcholine indicators Ach3.0 was injected into the DG of *ChAT-Cre* and *ChAT-Cre;Hrh1^{fl/fl}* mice and installed optic fibers above the DG for in vivo photometry recordings during the consolidation or retrieval phase of recent contextual fear memory (Fig. 4a and b)”

Lines 215-221 in revised manuscript – “By aligning the Ach signals with the video-annotated behavioral epochs in the retrieval phase of recent contextual fear memory, we observed behavior-related changes of Ach release across freezing or mobility bouts (Fig. 4e). During the retrieval phase of recent contextual fear memory, robust decreases in the Ach release usually occurred in the *ChAT-Cre* mice while slight Ach transients could be elicited in the *ChAT-Cre;Hrh1^{fl/fl}* mice before the onset of freezing bouts (Fig. 4f-h).”

Lines 231-234 in revised manuscript – “We next proved that whether MS^{ACh}-DG are functionally connected during the recent contextual fear retrieval phase, AAV-Ef1 α -DIO-axon-GCaMP6s was expressed in MS of *ChAT-Cre* and *ChAT-Cre;Hrh1^{fl/fl}* mice, and optical cannula were implanted in DG for fiber photometry (Fig. 4n-p).”

In short, Figs. 2a-d complicate the story line because the impact that these training-induced changes might have on memory retrieval in WT were not addressed.

One possibility is that those modest changes don't really have any impact in WT, allowing for proper modulation of Ach release during contextual memory. The authors imply instead that these changes will have an impact on memory, but that an unknown, hypothetical histaminergic input overrides them. I suggest they also consider the more parsimonious possibility above.

Response: Thank you for this valuable comment. We agree with reviewer's point. The decreased expression of H₁R will weaken the regulation of MS cholinergic neural excitability by histaminergic input during the contextual fear

retrieval. In this condition, we observed that *Hrh1* mRNA expression of MS cholinergic neurons significantly decreased (Fig. 2a and b) and their rheobase did not show statistical differences after 1 day of conditioning (Fig. 2c and d), indicating other inputs may also affect the excitability of MS cholinergic neurons and be involved in the regulation of contextual fear retrieval. Of note, our results showed that knocking down H₁R in MS cholinergic neurons results in the enhancement of contextual fear retrieval, at least suggesting the involvement of histaminergic system. Thus, further studies are needed to investigate the role of other neural inputs in the contextual fear retrieval. As suggested, we have revised our discussion. Please see lines 446-456.

Lines 446-456 in revised manuscript – “In addition, the decreased expression of H₁R will weaken the regulation of MS cholinergic neural excitability by histaminergic input during the contextual fear retrieval. In this condition, we found that although H₁R expression on MS cholinergic neurons significantly decreased after 1 day of conditioning session (Fig. 2a and b), their rheobase did not show statistical differences (Fig. 2c and d), indicating other inputs may also affect the excitability of MS cholinergic neurons and be involved in the regulation of contextual fear retrieval. Of note, our results showed that knocking down H₁R in MS cholinergic neurons results in the enhancement of contextual fear retrieval, at least suggesting the involvement of histaminergic system. Thus, further studies are needed to investigate the role of other neural inputs in the contextual fear retrieval.”

Reviewer #2 (Remarks to the Author):

The latest version of the manuscript is greatly improved and the authors have adequately addressed many of my concerns. There are only a few concerns / clarification remaining.

Response: We appreciate reviewer's positive and constructive comments, which have significantly helped us improve the manuscript. Our point-by-point responses are addressed below.

1) The addition of panels Fig3d,e helped to link the cFos histology with behavior, but the enhancement of context freezing in the *ChAT-Cre;Hrh1* mice did not reach statistical significance ($p = 0.053$). Either performing additional cFos and behavior experiments are required to achieve statistical significance or adding text to the Results section that refers to the lack of statistical significance in panels Fig3d,e is required.

Response: Thanks for the important suggestion. We have performed additional c-Fos and behavior experiments ($p = 0.0203$), and still found a significantly increased amount of c-Fos positive neurons in the DG of *ChAT-Cre;Hrh1^{fl/fl}* mice after contextual fear test (Fig 3 b-e in the revised manuscript), which was consistent with previous data (Fig 3 b-e in the original manuscript). The data has been added in revised manuscript as Fig. 3.

Revised Fig. 3b-e

Figure 3. The DG is the major downstream region of MS cholinergic neurons responsible for enhanced contextual fear memory in *ChAT-Cre;Hrh1^{fl/fl}* mice. (b) The percentage of c-Fos expressing neurons of the indicated areas. $n = 5-6$ mice per group. **(c)** The percentage of c-Fos expressing neurons in DG region of *ChAT-Cre* and *ChAT-Cre;Hrh1^{fl/fl}* mice re-exposed or not. $n = 3-6$ mice per group. **(d)** The curve of freezing level during the exploration period and each trial on the conditioning day for (a) and (b). $n = 6$ mice per group. **(e)** The curve of freezing level in each minute (left panel) and the percentage of freezing time during the contextual fear memory retrieval (right panel) for (a) and (b). $n = 6$ mice per group.

2) In the Methods or Results section, please specify the length of freezing bout considered for the DG and MS-DG fiber photometry quantification panels in Fig4. Providing a justification for using greater than 3 seconds in one set of analyses, but less than or greater than 3 sec in another set of analyses is required. Additionally, Fig4r-y appear to use a different data smoothing function compared to Fig4f-m. I was unable to find the details of smoothing parameters in the methods, please add these data processing details and justify if/why they were not applied similarly to the two fiber photometry data sets.

Response: We thank reviewer for the constructive comments.

1) The minimum length of freezing bout considered for the DG and MS-DG fiber photometry quantification panels in Fig4 is 2 s, which is consistent with the freezing criteria of fear conditioning task. The related method has been added in the revised manuscript. Please see method part, lines 601-602. We are sorry for the inappropriate symbols in Fig.4e and Fig.4q, since all the freezing bouts (all periods of inactivity with duration greater than 2 s) were considered in all analyses. According to reviewer's suggestion, the inappropriate symbols have been corrected in the revised manuscript.

Lines 601-602 in revised manuscript – “The minimum length of freezing bout considered for fiber photometry is 2 s, which is consistent with the freezing criteria of fear conditioning task.”

Revised Fig. 4e and q

(e) Example trace of ACh release in the DG and behavioral epochs during the contextual fear retrieval. Blue boxes above the traces indicate freezing bouts. Yellow boxes above the traces indicate mobility bouts. (q) Example trace of the fluorescence signal of MS projecting axons in DG and behavioral epochs during the contextual fear retrieval. Blue boxes above the traces indicate freezing bouts. Yellow boxes above the traces indicate mobility bouts.

2) We have checked the analyses conditions carefully, and confirmed that we didn't use any smoothing function for the data analyses in both Fig.4f-m and Fig.4r-y. We also consulted with professional technicians and obtained the same data. One possibility is that the viruses used in Fig.4f-m and Fig.4r-y are different (AAV-hSyn-ACh3.0 and AAV-Ef1 α -DIO-axon-GCaMP6s), which have different characteristics and kinetics. It is noteworthy that the similar data were observed in previous studies⁴⁻⁶, not just acetylcholine sensor. Following your constructive comment, we have added these data processing details in the revised manuscript. Please see method part, lines 594-601.

Lines 594-601 in revised manuscript – “Analysis code is available at GitHub repository: <https://github.com/wellsjay/TripplColorMultiChannelAnalysisPackage.git>. For the acquisition and analysis of all fiber photometry data, the parameter settings are consistent and unified. Time 0 was aligned to the onset of a behavioral epoch. F0 is the baseline average fluorescence signals of the 2 seconds before Time 0. The fluorescence responses were indicated by $\Delta F/F$ (calculated as $(F-F_0)/F_0$). The area under the curve of the $\Delta F/F$ plot was measured to quantify the response to contextual fear retrieval in the DG.”

3) Regarding freezing criteria for PANLAB software, please specify the minimum duration for immobilization to be considered freezing.

Response: Thank you for this valuable comment. The minimum duration for immobilization to be considered freezing is 2 s, and has been added in the revised manuscript. Please see method part, lines 500-502.

Lines 500-502 in revised manuscript – “The minimum duration for immobilization to be considered freezing is 2 s. All periods of inactivity with duration lower than 2 s will not be taken into account.”

Reference

- 1 Boyce, V. S., Park, J., Gage, F. H. & Mendell, L. M. Differential effects of brain-derived neurotrophic factor and neurotrophin-3 on hindlimb function in paraplegic rats. *Eur J Neurosci* **35**, 221-232, doi:10.1111/j.1460-9568.2011.07950.x (2012).
- 2 Laplante, F., Morin, Y., Quirion, R. & Vaucher, E. Acetylcholine release is elicited in the visual cortex, but not in the prefrontal cortex, by patterned visual stimulation: a dual in vivo microdialysis study with functional correlates in the rat brain. *Neuroscience* **132**, 501-510, doi:10.1016/j.neuroscience.2004.11.059 (2005).
- 3 Decker, A. L. & Duncan, K. Acetylcholine and the complex interdependence of memory and attention. *Current Opinion in Behavioral Sciences* **32**, 21-28, doi:https://doi.org/10.1016/j.cobeha.2020.01.013 (2020).
- 4 Sun, Q. *et al.* Acetylcholine deficiency disrupts extratelencephalic projection neurons in the prefrontal cortex in a mouse model of Alzheimer's disease. *Nat Commun* **13**, 998,

doi:10.1038/s41467-022-28493-4 (2022).

- 5 Ji, Y. W. *et al.* Plasticity in ventral pallidal cholinergic neuron-derived circuits contributes to comorbid chronic pain-like and depression-like behaviour in male mice. *Nat Commun* **14**, 2182, doi:10.1038/s41467-023-37968-x (2023).
- 6 Wang, X. Y. *et al.* A glutamatergic DRN-VTA pathway modulates neuropathic pain and comorbid anhedonia-like behavior in mice. *Nat Commun* **14**, 5124, doi:10.1038/s41467-023-40860-3 (2023).

REVIEWERS' COMMENTS

Reviewer #1 (Remarks to the Author):

The authors have addressed previous concerns.

Reviewer #2 (Remarks to the Author):

The authors have adequately addressed my concerns.

Reviewer #2 (Remarks on code availability):

I did not run the code, but the Github page provides a README file and the instructions and commenting within the .m files appears appropriate to enable reproduction of results.

Point-by-point response to reviewers' comments

REVIEWERS' COMMENTS

Reviewer #1 (Remarks to the Author):

The authors have addressed previous concerns.

Response: We appreciate the positive feedback from the reviewer. We would like to thank the reviewer for the comments which have helped us strengthen the manuscript.

Reviewer #2 (Remarks to the Author):

The authors have adequately addressed my concerns.

Response: We are grateful to reviewer for the valuable suggestions that have helped us improve our manuscript.

Reviewer #2 (Remarks on code availability):

I did not run the code, but the Github page provides a README file and the instructions and commenting within the .m files appears appropriate to enable reproduction of results.

Response: We thank the reviewer again for taking the time to review our manuscript and have provided us these constructive suggestions.